# Theory of Autoregressive Diffusion Model: Inference Complexity and Conditional Dependency Learning

## Abstract

Autoregressive (AR) diffusion models have recently attracted significant attention for their ability to generate high-quality, diverse samples across various tasks involving text, image, and video generation. Despite this surge of interest, the theoretical underpinnings of AR diffusion remain largely unexplored. This work, for the first time, investigates the inference complexity and underlying mechanisms behind AR diffusion's strong performance. Building on the sequential patch-by-patch generation paradigm, we formalize the inference process as a series of stage-wise conditional distribution samplings. This formulation yields that, when conditional components are learned accurately, the resulting approximation to the full joint distribution becomes highly precise. Our theoretical analysis establishes the AR diffusion inference complexity bound for a general number of stages $K$, requiring only minimal smoothness assumptions on the score functions and their estimation error. The complexity includes an additional factor proportional to the number of stages, reflecting the model's sequential architecture. On the other hand, we show that this stage-wise design can be advantageous for learning specific conditional dependencies between patches, which may be overlooked by conventional diffusion models that focus primarily on joint distributions. Subsequent experiments on synthetic data validate this theoretical insight.

## 1 Introduction

Autoregressive (AR) diffusion models have recently garnered considerable attention in various domains, including image generation (Li et al., 2024d; Sun et al., 2024; Tian et al., 2024a), video synthesis (Weng et al., 2023; Liu et al., 2024; Sun et al., 2025), and text generation (Wu et al., 2023; Tian et al., 2024b). AR diffusion can be viewed as a hybrid approach combines techniques from autoregressive modeling and diffusion-based methods (Dhariwal & Nichol, 2021; Austin et al., 2021; Ramesh et al., 2022; Saharia et al., 2022; Gupta et al., 2024; Luo et al., 2023). Specially, AR diffusion generates image patches sequentially according to a predetermined (e.g., raster-scan) order, while employing standard diffusion methods to approximate distribution of each patch.

Alongside the rapid advancement of diffusion-based applications, diffusion models' corresponding theoretical foundations have also made significant progress Chen et al. (2023b;a); Benton et al. (2024); Li & Yan (2025). The sampling error of diffusion models primarily stems from three sources: (1) initialization error arising from the distributional mismatch between the terminal distribution after forward noising, and the initial distribution of the reverse process; (2) score estimation error representing the discrepancy between learned and true score; and (3) discretization error from approximating continuous sampling dynamics with discrete-time steps. Recent work (Chen et al., 2023b; Lee et al., 2022) demonstrate that with accurate score estimation, diffusion models can achieve polynomial-time convergence under minimal smoothness assumptions, without requiring structural conditions like log-concavity on the data distribution.

Compared to diffusion models, while AR diffusion models have been applied with great success in various domains (Li et al., 2024d; Sun et al., 2024; Tian et al., 2024a; Wu et al., 2023; Tian et al., 2024b; Weng et al., 2023; Liu et al., 2024; Sun et al., 2025), their theoretical foundations remain comparatively underexplored. In this paper, we aim to narrow this gap by investigating

the theoretical properties of AR diffusion models, examining in particular *whether they can offer convergence guarantees under the general assumptions commonly adopted by standard diffusion models*. In addition, AR diffusion is designed to learns conditional probability distributions along a predefined order (e.g., raster scan order), whereas standard diffusion models focus on modeling the joint distribution. An intuitive consideration is that when data exhibit dependencies, such as linear constraints among image patches (Han et al., 2025b; Lu et al., 2025; Wang et al., 2025), *AR diffusion, by emphasizing conditional probabilities, can more effectively capture such dependencies*. We aim to provide theoretical insights into the above two questions.

To address these questions, we present the first theoretical analysis of the inference complexity of AR diffusion models. We begin by interpreting the existing AR inference process as the reverse of a stage-wise Ornstein–Uhlenbeck (OU) process, where the initial distribution coincides with a particular conditional data distribution. Compared to convergence analysis in conventional diffusion, two key challenges emerge in AR diffusion: (1) The global training loss only minimizes the average diffusion loss across stages Li et al. (2024c), providing no guarantee of uniformly small score estimation error for each stage. We must ensure this global objective can effectively mitigate error accumulation during multi-stage inference; and (2) extending the second-moment and smoothness assumptions from standard diffusion analysis to every conditional distribution—a requirement that may be too restrictive in AR settings. We resolve these issues by developing a more refined characterization of the sampling error propagation in AR diffusion models and show that, under appropriately bounded score estimation errors, the total number of score evaluations needed to achieve an $\epsilon$-accurate sample in terms of KL divergence is on the order of $\tilde{O}(K\,d\,\epsilon^{-2})$. Beyond the general regime, we further conduct the comparison between AR diffusion and standard diffusion models through the lens of capturing the conditional dependencies among patches. Intuitively, the stage-wise learning of AR diffusion may inherently promote the recovery of conditional distributions, whereas standard diffusion models focus primarily on the joint distribution. This suggests a potential advantage for AR diffusion in learning certain fine-grained details of the data distribution.

**Contributions.** We provide the first rigorous theoretical analysis of AR diffusion models. Using a stage-wise forward Ornstein-Uhlenbeck (OU) process, we show that: (1) the global AR diffusion objective, derived from local denoising score entropy, matches the expected loss in empirical studies Li et al. (2024c), and (2) it effectively controls accumulation of inference errors. Under assumptions nearly as mild as those for standard diffusion (Benton et al., 2024), our analysis establishes that AR diffusion requires a gradient complexity of $\tilde{O}(K\,d\,\epsilon^{-2})$ to achieve $O(\epsilon)$ sampling error, provided the score estimation error is at most $O(\epsilon/\sqrt{K})$, a result consistent with empirical observations (i.e., smaller training loss yet lower efficiency compared to conventional diffusion). Moreover, we prove that, in certain regimes , AR diffusion can better capture some conditional dependencies among patches compared to standard diffusion. We show that the AR diffusion model can always achieve a vanishing sampling error bound for conditional distributions, whereas standard diffusion may suffer from an *unbounded* KL divergence error in these conditionals, even when its error in the joint distribution remains well-controlled. Numerical experiments on synthetic data corroborate these theoretical findings.

## 2 PRELIMINARIES

In this section, we first introduce the notation used in subsequent sections and then adapt the algorithm presented in Li et al. (2024c) to our chosen notation system.

**Conditional Distribution Decomposition.** Suppose we divide the vector $\boldsymbol{x} \in \mathbb{R}^d$ into $K$ patches $(\boldsymbol{x}_1, \boldsymbol{x}_2, \ldots, \boldsymbol{x}_K)$ following predefined order (e.g., raster scan); thus, a general distribution on $\mathbb{R}^d$ can be viewed as a joint distribution $p(\boldsymbol{x}_1, \boldsymbol{x}_2, \ldots, \boldsymbol{x}_K)$ where $\boldsymbol{x}_k \in \mathbb{R}^{d_k}$ and $\sum_{k=1}^{K} d_k = d$. Given an index set $S$, it can deduce a vector $\boldsymbol{x}_S := (\boldsymbol{x}_i)_{i \in S}$ from the joint $\boldsymbol{x}$, and a distribution also from a joint distribution:

$$p_S(\boldsymbol{x}_S) := \int_{(\boldsymbol{x}_i)_{i \notin S}} p(\boldsymbol{x}_1, \boldsymbol{x}_2, \ldots, \boldsymbol{x}_K)\, \mathrm{d}((\boldsymbol{x}_i)_{i \notin S}).$$

In what follows, we focus on sets with consecutive indexes denoted as $[l:r] = \{l, l+1, \ldots, r\}$ and deduce the corresponding distribution

$$p_{[l:r]}(\boldsymbol{x}_{[l:r]}) = p_{[l:r]}(\boldsymbol{x}_l, \boldsymbol{x}_2, \ldots, \boldsymbol{x}_r) = \int_{\boldsymbol{x}_{[1:l-1]}, \boldsymbol{x}_{[r+1,K]}} p(\boldsymbol{x}_{[1:K]})\, \mathrm{d}(\boldsymbol{x}_{[1:l-1]}, \boldsymbol{x}_{[r+1,K]}). \quad (1)$$

If $k \notin [l:r]$ and $(\boldsymbol{x}_l, \ldots, \boldsymbol{x}_r)$ is given, the conditional probability on $\mathbf{x}_k$ is defined as

$$p_{k|[l:r]}(\boldsymbol{x}_k | \boldsymbol{x}_l, \ldots, \boldsymbol{x}_r) = \frac{p_{[l:r] \cup \{k\}}(\boldsymbol{x}_{[l:r]}, \boldsymbol{x}_k)}{p_{[l:r]}(\boldsymbol{x}_{[l:r]})}. \tag{2}$$

In addition, the density function of the Gaussian-type distribution $\mathcal{N}(\mathbf{0}, \sigma^2 \boldsymbol{I})$ is abbreviated as $\varphi_{\sigma^2}$.

After the general definition, we consider some notations related to diffusion models. Specifically, the data density function is denoted as $p_* \propto \exp(-f_*) \colon \mathbb{R}^{d_1 + d_2 + \ldots + d_K} \to \mathbb{R}$, so the marginal and conditional distributions derived from any index set are $p_{*,S}$ and $p_{*,k|[l:r]}$.

**AR Diffusion Models.** Following Li et al. (2024c), we briefly revisit AR diffusion models, which usually divide the generated data into several patches, e.g., $\hat{\mathbf{x}} = [\hat{\mathbf{x}}_1, \hat{\mathbf{x}}_2, \ldots, \hat{\mathbf{x}}_K]$ and then predict the next patch, e.g., $\hat{\mathbf{x}}_{k+1}$ in a sequence based on the previous ones e.g., $\hat{\boldsymbol{x}}_{1:k}$ or a compressed representation, e.g., $\boldsymbol{z} \coloneqq \boldsymbol{g}_{\boldsymbol{\theta}_{\mathrm{ar}}}(\hat{\boldsymbol{x}}_{1:k})$. The prediction is usually to draw $\hat{\mathbf{x}}_{k+1}$ from a specific distribution $q_0(\cdot | \boldsymbol{x}_{[1:k]})$ inspired by typical diffusion models. Then, considering a OU process initialized by $q_0(\cdot | \boldsymbol{x}_{[1:k]})$, i.e.,

$$\mathrm{d}\mathbf{y}_t = -\mathbf{y}_t \mathrm{d}t + \sqrt{2}\mathrm{d}\boldsymbol{B}_t, \ \mathbf{y}_t \sim q_t(\cdot | \boldsymbol{x}_{[1:k]}), \ t \in (0, T], \tag{3}$$

drawing sample from $q_0(\cdot | \boldsymbol{x}_{[1:k]})$ is equivalent to obtain $\mathbf{y}_0$ by reversing SDE. 3 and run

$$\mathrm{d}\mathbf{y}_t^{\leftarrow} = \left(\mathbf{y}_t^{\leftarrow} + 2\nabla \ln q_{T-t}(\mathbf{y}_t^{\leftarrow} | \boldsymbol{x}_{[1:k]})\right) \mathrm{d}t + \sqrt{2}\mathrm{d}\boldsymbol{B}_t, \tag{4}$$

where $\mathbf{y}_t^{\leftarrow}$ follows distribution $q_t^{\leftarrow} = q_{T-t}$. Previous work approximately solves the above SDE as

$$\mathrm{d}\hat{\mathbf{y}}_t = \left(\hat{\mathbf{y}}_t + \boldsymbol{s}_{\boldsymbol{\theta}_{\mathrm{dm},k+1}}(\hat{\mathbf{y}}_{t_r} | T - t_r, \boldsymbol{z})\right) \mathrm{d}t + \sqrt{2}\mathrm{d}\boldsymbol{B}_t. \tag{5}$$

In this formulation, the score estimator $\boldsymbol{s}_{\boldsymbol{\theta}_{\mathrm{dm},k+1}}(\cdot | T - t_r, \boldsymbol{z})$ depending on $\boldsymbol{z}$ is parameterized by $\boldsymbol{\theta}_{\mathrm{dm},k+1}$ and used to approximate $\nabla \ln q_{T-t_r}(\cdot | \boldsymbol{x}_{[1:k]})$. Besides, $t_r$ denotes the timestamps belonging to the set $\{t_r\}_{r=0}^R$, which partitions the mixing time $T$ of the forward process $\{\mathbf{y}_t\}_{t=0}^T$ into $R$ segments of lengths $\{\eta_r\}_{r=0}^{R-1}$. Given these definitions, if we set

$$\xi_{\boldsymbol{\theta}}(\boldsymbol{y} | t, \boldsymbol{z}) \coloneqq -(1 - e^{-2t})^{-1/2} \cdot \boldsymbol{s}_{\boldsymbol{\theta}}(\boldsymbol{y} | t, \boldsymbol{z}),$$

and the loss at $T - t$ to be

$$L_{k+1,t}(\boldsymbol{\theta}_{\mathrm{dm},k+1}, \boldsymbol{\theta}_{\mathrm{ar}} | \boldsymbol{x}_{[1:k]}) = \mathbb{E}_{\mathbf{y}_0 \sim q_0(\cdot | \boldsymbol{x}_{[1:k]}), \xi \sim \mathcal{N}(\mathbf{0}, \boldsymbol{I})} \left[ \left\| \xi - \xi_{\boldsymbol{\theta}_{\mathrm{df}+1,k}}(\mathbf{y}_{T-t} | T - t, \ g_{\boldsymbol{\theta}_{\mathrm{ar}}}(\boldsymbol{x}_{[1:k]})) \right\|^2 \right],$$

then Li et al. (2024c) proposes the following objective for training $\boldsymbol{\theta}_{\mathrm{dm},k+1}$ and $\boldsymbol{\theta}_{\mathrm{ar}}$:

$$L(\boldsymbol{\theta}_{\mathrm{dm},k+1}, \boldsymbol{\theta}_{\mathrm{ar}} | \boldsymbol{x}_{[1:k]}) = \mathbb{E}_t \left[ L_{k+1,t}(\boldsymbol{\theta}_{\mathrm{dm},k+1}, \boldsymbol{\theta}_{\mathrm{ar}} | \boldsymbol{x}_{[1:k]}) \right]. \tag{6}$$

In the subsequent discussion, for simplicity, we do not explicitly distinguish different learnable parameters (e.g., $\boldsymbol{\theta}_{\mathrm{dm},1}, \ldots, \boldsymbol{\theta}_{\mathrm{dm},K}, \boldsymbol{\theta}_{\mathrm{ar}}$) and use $\boldsymbol{\theta}$ to represent all relevant parameters instead. Moreover, in handling SDEs within Alg. 2 for some fixed $k$ and $\boldsymbol{z} = g_{\boldsymbol{\theta}_{\mathrm{ar}}}(\boldsymbol{x}_{[1:k]})$, we abbreviate the underlying distribution $q_t(\cdot | \boldsymbol{x}_{[1:k]})$ by $q_t(\cdot)$.

**General Assumptions.** To study convergence and the gradient complexity required for achieving small total variation (TV) distance or Kullback–Leibler (KL) divergence, we assume $p_*$ satisfies:

**[A1]** The second moment of $p_*$ is bounded, i.e.,

$$\mathbb{E}_{\mathbf{x} \sim p_*}[\|\mathbf{x}\|^2] = \int p_*(\boldsymbol{x}) \|\boldsymbol{x}\|^2 \, \mathrm{d}\boldsymbol{x} \leq m_0.$$

**[A2]** The energy function of $p_*$ has a bounded Hessian and bounded gradient, namely

$$\|\nabla^2 \ln p_*\| \leq L \quad \text{and} \quad \|\nabla \ln p_*\| \leq \sqrt{L}.$$

Assumption **[A1]** is prevalent across most works on diffusion analysis. Assumption **[A2]** only imposes the Hessian upper bound on the data distribution's energy function, which is often often referred to as a minimal smoothness requirement Chen et al. (2023a). Although, compared with previous works, an additional gradient norm upper bound is only required, it does not have any constraint on the isoperimetric property, which means the data distribution is still allowed to be general non-log-concave.

---

**Algorithm 1** SIMPLIFIED AUTOREGRESSIVE DIFFUSION GENERATION

---

1: **Input:** Number of patches $K$, mixing time $T$ for each patch, number of iterations $R$ per patch, score estimator $s_{\boldsymbol{\theta}}$, condition generator $g_{\boldsymbol{\theta}_{\mathrm{ar}}}$
2: For the mixing time $T$, define two sequences:

$$\{t_r\}_{r=0}^R, \quad t_0 = 0, \quad t_R = T, \quad t_i \le t_j \ \forall \ i \le j,$$
$$\{\eta_r\}_{r=0}^{R-1}, \quad \eta_r = t_{r+1} - t_r. \tag{7}$$

3: Generate $\hat{\boldsymbol{x}}_1$ by calling Alg. 2$(0, \emptyset, R, \{t_r\}_{r=0}^R, s_\theta)$.
4: **for** $k = 1$ to $K - 1$ **do**
5:    Acquire the condition for the next patch:

$$\boldsymbol{z} := g_{\boldsymbol{\theta}_{\mathrm{ar}}}(\hat{\boldsymbol{x}}_{[1:k]}) = g_{\boldsymbol{\theta}_{\mathrm{ar}}}(\hat{\boldsymbol{x}}_1, \hat{\boldsymbol{x}}_2, \ldots, \hat{\boldsymbol{x}}_k). \tag{8}$$

6:    Generate the next patch $\hat{\boldsymbol{x}}_{k+1}$ conditioned on $\boldsymbol{z}$ by calling Alg. 2$(k, \boldsymbol{z}, R, \{t_r\}_{r=0}^R, s_\theta)$.
7: **end for**
8: **return** The concatenated patches $[\hat{\boldsymbol{x}}_1, \hat{\boldsymbol{x}}_2, \ldots, \hat{\boldsymbol{x}}_K]$.

---

## 3 STAGE-WISE FORMULATION OF AR DIFFUSION

In this section, we formalize the AR diffusion framework from a theoretical perspective by decomposing it into two stages: an autoregressive stage, where the next patch is predicted, and a diffusion stage, where the inference for each patch is viewed as the reverse of a stage-wise OU process. This stage-wise theoretical formulation provides the necessary foundation for the convergence results presented in Section 4.

**Understanding of the inference.** In Algorithm 1, we summarize the inference procedure for AR diffusion where Step 4-7 show the most essential characteristics of AR diffusion. At each iteration, we generate the next patch, $\hat{\boldsymbol{x}}_{k+1}$, by conditioning on the previously generated patches, $\hat{\boldsymbol{x}}_{[1:k]}$, or their compressed representations, $\boldsymbol{z} = g_{\boldsymbol{\theta}_{\mathrm{ar}}}(\hat{\boldsymbol{x}}_{[1:k]})$. The specific process for sampling $\hat{\boldsymbol{x}}_{k+1}$, shown in Algorithm 2, is analogous to typical diffusion-based inference that aims to recover $q_0(\cdot|\hat{\boldsymbol{x}}_{[1:k]})$. However, Algorithm 2 distinguishes itself by introducing a non-uniform partition $\{t_r\}_{r=0}^R$ in Eq. 7 for the reverse process (Eq. 4), thereby offering additional flexibility compared to standard diffusion approaches. Under this condition, we argue that such a procedure can be understood as the reverse of a stage-wise OU process by setting the initial distribution of each stage to satisfy

$$q_0(\cdot|\boldsymbol{x}_{[1:k]}) = p_{*,k+1|[1:k]}(\cdot|\boldsymbol{x}_{[1:k]}). \tag{9}$$

Specifically, the forward OU process with $K$ (patch number) stages can be described as follows

1. For Stage $k$, we consider a random process initialized by the distribution

$$p_{*,[1:K-k+1]} = p_{*,[1:K-k]} \cdot p_{*,K-k+1|[1:K-k]}.$$

2. Given the random variable $\mathbf{y}_t$ implemented as in Eq. 3, we have

$$\{[\mathbf{x}_{[1:K-k]}, \mathbf{y}_t]\} \sim p_{*,[1:K-k]} \cdot q_t \quad \text{where} \quad q_0 = p_{*,K-k+1|[1:K-k]}.$$

3. Since $q_T \to \mathcal{N}(\mathbf{0}, \boldsymbol{I})$ as $T \to \infty$, we can approximate the underlying distribution of $\{[\mathbf{x}_{[1:K-k]}, \mathbf{y}_T]\}$ by

$$p_{*,[1:K-k]} \cdot q_T \approx p_{*,[1:K-k]} \cdot \varphi_1.$$

After $K$ recursions, the entire forward process converges to a product of standard Gaussians with different dimensions. Correspondingly, the underlying mechanism of Alg. 1 proceeds as follows.

1. For Stage $k$, suppose we have sample $\hat{\mathbf{x}}_{1:k-1}$ with underlying distribution $\hat{p}_{[1:K-1]} \approx p_{*,[1:K-1]}$.
2. Alg. 2 can approximately reverse Eq. 3 and obtain a random variable $\hat{\mathbf{y}}$ satisfying

$$\hat{\mathbf{y}} \sim \hat{p}_{*,K|K-1}(\cdot|\hat{\mathbf{x}}_{1:k-1}) \approx p_{*,k|[1:k-1]}(\cdot|\hat{\boldsymbol{x}}_{[1:k-1]})$$

---

**Algorithm 2** PATCH INFERENCE PROCESS UNDER GIVEN CONDITIONS

---

1: **Input:** Patch index $k$, latent vector $\boldsymbol{z}$ (conditions), number of iterations $R$, time sequence $\{t_r\}_{r=0}^R$, score estimator $\boldsymbol{s_\theta}$
2: Draw an initial sample $\hat{\mathbf{y}}_0 \sim \mathcal{N}(\mathbf{0}, \boldsymbol{I}_{d_{k+1}})$.
3: **for** $r = R - 1$ **downto** $0$ **do**
4:     Suppose $\xi \sim \mathcal{N}(\mathbf{0}, \boldsymbol{I})$, use an exponential integrator to simulate the SDE:

$$\hat{\mathbf{y}}_{t_{r+1}} = e^{\eta_r}\hat{\mathbf{y}}_{t_r} + (e^{\eta_r} - 1) \cdot 2\boldsymbol{s}_{\boldsymbol{\theta}_{\mathrm{df},k+1}}(\hat{\mathbf{y}}_{t_r} \mid t_R - t_r, \boldsymbol{z}) + \sqrt{e^{2\eta_r} - 1} \cdot \xi \qquad (10)$$

5: **end for**
6: **return** $\hat{\mathbf{y}}_{t_R}$.

---

3. Concat all variables, i.e., $\hat{\mathbf{x}}_{[1:k]} = [\hat{\mathbf{x}}_{1:k-1}, \hat{\mathbf{y}}]$ and use it as the conditioning of the next stage.

**Derivation of the global training objective.** To implement Alg. 2, the core step Eq. 10 is based on the well-trained neural score estimator, i.e., $\boldsymbol{s}_{\boldsymbol{\theta}_{\mathrm{df},k+1}}(\cdot \mid t_R - t_r, \boldsymbol{z})$ for any $k$ and $r$. Following from the stage-wise forward and reverse process mentioned before, we deduce a global objective accounting for distributions over $k$, $\boldsymbol{z}$ (or $\boldsymbol{x}_{[1:k]}$), $t$, and $\mathbf{y}_0$ from Eq. 6.

- A convenient choice for the distribution of $k$ is uniform sampling from $\{1, 2, \ldots, K\}$.

- To estimate the expectation of $L(\boldsymbol{\theta}_{\mathrm{dm},k+1}, \boldsymbol{\theta}_{\mathrm{ar}} | \mathbf{x}_{[1:k]})$ with the random variable $\boldsymbol{z}$ (or $\mathbf{x}_{[1:k]}$), we can let $\mathbf{x}_{[1:k]} \sim p_{*,[1:k]}$. In practice, the underlying distribution can be approximated by

$$p_{*,[1:k]}(\boldsymbol{x}_{[1:k]}) \approx \frac{1}{U}\sum_{i=1}^{U} \delta_{\boldsymbol{u}_{[1:k]}^{(i)}}(\boldsymbol{x}_{[1:k]}), \qquad (11)$$

where $\boldsymbol{u}^{(i)}$ is a ground-truth sample from the training set of size $U$, and $\delta$ denotes a Dirac measure.

- To build $L_{k+1,t}(\boldsymbol{\theta}_{\mathrm{dm},k+1}, \boldsymbol{\theta}_{\mathrm{ar}} \mid \boldsymbol{x}_{[1:k]})$ in Eq. 6, we need samples $\mathbf{y}_t$ for each $t \in [0, T]$. Since $\{\mathbf{y}_t\}_{t=0}^T$ follows an OU process, it suffices to draw $\mathbf{y}_0$ when we set $q_0(\cdot|\boldsymbol{x}_{[1:k]}) = p_{*,k+1|[1:k]}(\cdot|\boldsymbol{x}_{[1:k]})$ and approximate RHS of the equation with

$$p_{*,k+1|[1:k]}(\boldsymbol{x}_{k+1}|\boldsymbol{x}_{[1:k]}) \approx \frac{\sum_{i=1}^{U} \delta_{\boldsymbol{u}_{[1:k+1]}^{(i)}}(\boldsymbol{x}_{[1:k]}, \boldsymbol{x}_{k+1})}{\sum_{i=1}^{U} \delta_{\boldsymbol{u}_{[1:k]}^{(i)}}(\boldsymbol{x}_{[1:k]})}.$$

- We set the distribution of $t$ to be uniform over the set $\{T - t_0, T - t_1, \ldots, T - t_{R-1}\}$ for ease of implementation.

Under these settings, let the conditional denoising score-matching loss be written as

$$L_{k+1,r}^{\mathrm{DSM}}(\boldsymbol{\theta}|\boldsymbol{x}_{[1:k]}) \coloneqq \mathbb{E}_{\mathbf{y}_0 \sim p_{*,k+1|[1:k]}(\cdot|\boldsymbol{x}_{[1:k]}), \xi \sim \mathcal{N}(\mathbf{0},\boldsymbol{I})}\left[\left\|\xi - \xi_{\boldsymbol{\theta}}\left(\mathbf{y}_{T-t_r} \mid T - t_r, \ g_{\boldsymbol{\theta}}(\boldsymbol{x}_{[1:k]})\right)\right\|^2\right], \qquad (12)$$

then the global objective becomes

$$L^{\mathrm{DSM}}(\boldsymbol{\theta}) \coloneqq \frac{1}{KR}\sum_{k=1}^{K}\sum_{r=0}^{R-1} \mathbb{E}_{\mathbf{x}_{[1:K]} \sim p_*}\left[L_{k,t_r}^{\mathrm{DSM}}(\boldsymbol{\theta}|\mathbf{x}_{[1:k-1]})\right]. \qquad (13)$$

Here, we slightly abuse notation because $\boldsymbol{x}_{[1:0]}$ is undefined. In fact, Alg. 1 shows that generating $\mathbf{x}_1$ is unconditional, so we do not need $p_{[1:0]}$, and $p_{*,1|[1:0]}$ only needs to match $p_{*,1}$. Actually, Eq. 12 and Eq. 13 are formulated to implement the training by fitting the noise. While in analysis, the conditional score-matching loss formulated as follows

$$L_{k+1,r}^{\mathrm{SM}}(\boldsymbol{\theta}|\boldsymbol{x}_{[1:k]}) = \mathbb{E}_{\mathbf{y}_0 \sim p_{*,k+1|[1:k]}(\cdot|\boldsymbol{x}_{[1:k]}), \xi \sim \mathcal{N}(\mathbf{0},\boldsymbol{I})}\left[\left\|\boldsymbol{s_\theta}\left(\mathbf{y}' \mid T - t_r, g_{\boldsymbol{\theta}}(\boldsymbol{x}_{[1:k]})\right) - \nabla \ln q_{T-t_r}\left(\mathbf{y}' \mid \boldsymbol{x}_{[1:k]}\right)\right\|^2\right],$$

will be more concerned about where $\mathbf{y}'$ satisfies

$$\mathbf{y}' = e^{-(T-t_r)} \cdot \mathbf{y}_0 + \sqrt{1 - e^{-2(T-t_r)}} \cdot \xi.$$

Corresponding to Eq. 13, we consider a global score-matching loss formulated as

$$L^{\mathrm{SM}}(\boldsymbol{\theta}) = \frac{1}{KR} \sum_{k=1}^{K} \sum_{r=0}^{R-1} \mathbb{E}_{\mathbf{x}_{[1:k-1]} \sim p_{*,[1:k-1]}} \cdot \left[ (1 - e^{-2(T-t_r)}) \cdot L_{k,r}^{\mathrm{SM}}(\boldsymbol{\theta}|\mathbf{x}_{[1:k-1]}) \right]. \tag{14}$$

Compared with with Eq. 13, we may note the weight of $L_{k,r}^{\mathrm{SM}}(\boldsymbol{\theta}|\mathbf{x}_{[1:k-1]})$ is not uniformed. While the additional factor $(1 - e^{-2(T-t_r)})$ will be canceled by a different choice of $t$'s distribution. For example, we can sample from $\{\hat{t}_r\}$ defined as

$$\{\hat{t}_r\}_{r=0}^{R} \coloneqq \{T - t_r\}_{r=0}^{R} \quad \text{with} \quad \Pr(r) = \frac{1 - e^{-2\hat{t}_r}}{\sum_{\tau=0}^{R}(1 - e^{-2\hat{t}_\tau})}.$$

Under these settings, AR diffusions follows the lemma below (proof is deferred to Appendix A).

**Lemma 1.** *For any $\boldsymbol{\theta} \in \mathrm{dom}(L^{\mathrm{DSM}})$, it holds that*

$$\nabla_{\boldsymbol{\theta}} L^{\mathrm{DSM}}(\boldsymbol{\theta}) = \nabla_{\boldsymbol{\theta}} L^{\mathrm{SM}}(\boldsymbol{\theta}).$$

**Remark 1.** *This lemma explicitly shows that minimizing the global objective and the global score matching in AR diffusion with a gradient-based optimizer is equivalent. When the objective, i.e., Eq. 13, is well optimized, we can expect to have a highly accurate score estimation. Then, it is reasonable for us to propose the following assumption.*

**[A3]** The score training error satisfies

$$\frac{1}{KR} \sum_{k=1}^{K} \sum_{r=0}^{R-1} \mathbb{E}_{p_{*,[1:k-1]}} \left[ L_{k,r}^{\mathrm{SM}}(\boldsymbol{\theta}|\mathbf{x}_{[1:k-1]}) \right] \leq \epsilon_{\mathrm{score}}^2.$$

**Initial distribution requirements for inference convergence.** Consider that AR diffusion models use $p_{*,k+1|[1:k]}(\cdot|\boldsymbol{x}_{[1:k]})$ as the initial distribution at each stage of forward process, we expect $p_{*,k+1|[1:k]}(\cdot|\boldsymbol{x}_{[1:k]})$ to exhibit the same theoretical properties typically assumed for data distributions in standard diffusion models.

**Lemma 2.** *Suppose Assumption [A1] holds. Then, for any $K > k' \geq 1$, we have*

$$\sum_{k=0}^{k'} \mathbb{E}_{\mathbf{x}_{[1:k]} \sim p_{*,[1:k]}} \left[ \mathbb{E}_{\mathbf{y} \sim p_{*,k+1|[1:k]}(\cdot|\mathbf{x}_{[1:k]})} \left[ \|\mathbf{y}\|^2 \right] \right] \leq m_0 \quad and \quad \mathbb{E}_{\mathbf{x}_{[1:k']} \sim p_{*,[1:k']}} [\|\mathbf{x}_{1:k'}\|^2] \leq m_0.$$

**Lemma 3.** *Suppose Assumption [A2] holds. For any $k > 1$, any $\boldsymbol{x}, \boldsymbol{x}' \in \mathbb{R}^{d_k}$, and any $\boldsymbol{y} \in \mathbb{R}^{d_1+d_2+\ldots+d_{k-1}}$, we have*

$$\left\| \nabla \ln \frac{p_{*,k|[1:k-1]}(\boldsymbol{x}|\boldsymbol{y})}{p_{*,[k|[1:k-1]]}(\boldsymbol{x}'|\boldsymbol{y})} \right\| \leq 2L \|\boldsymbol{x} - \boldsymbol{x}'\|.$$

*Moreover, we have $\|\nabla^2 \ln p_{*,[1:1]}(\cdot)\| \leq 2L$.*

**Remark 2.** *Compared with the second-moment bound assumed in typical diffusion analyses (e.g., Chen et al. (2023b;a)), there is no uniform second-moment bound on the initial distributions for all stages in the AR diffusion setting. Hence, we require adaptive convergence for different $p_{*,k+1|[1:k]}$, then removing particle dependence by taking the expectation. Moreover, the score smoothness condition (Lemma 3) is satisfied by the initial distributions for all stages, which aligns with the score smoothness requirements on data distributions in Chen et al. (2023a).*

## 4 THEORETICAL GUARANTEES FOR AR DIFFUSION

In this section, we provide the theoretical results of the AR diffusion models. We first analyze in general the inference and training performance of AR diffusion. We demonstrate that, compared with typical DDPM, its gradient complexity increases by a factor of $K$ (the number of data patches) during the inference time, but it is practical for large-scale applications. We then theoretically analyze AR diffusion's property for capturing conditional dependence structures within data, formally demonstrating its advantages in learning feature dependencies compared to typical diffusion models.

## 4.1 INFERENCE PERFORMANCES OF AR DIFFUSION

Building upon Lemma 4, we deliver the theoretical result on the sampling error of AR diffusion model on the joint distributions over all the random variables $\boldsymbol{x}_{[1:K]}$.

**Theorem 1.** *Suppose Assumption [A1]-[A3] hold, and $\delta \leq \left(0, \ln \sqrt{(4L)^{-2} + 1} + (4L)^{-1}\right]$, if Alg. 1 chooses the time sequence $\{\eta_r\}_{r=0}^{R-1}$ as*

$$\eta_r = \begin{cases} \eta & when \quad 0 \leq r < M \\ \eta/(1+\eta)^{r-M+1} & when \quad M \leq r < N \\ \eta & when \quad N \leq r \leq R \end{cases}$$

*where*

$$M = \frac{T-1}{\eta}, \quad N = M + \frac{2\ln(1/\delta)}{\eta}, \quad and \quad R = N + \frac{\delta}{\eta},$$

*then, the generated samples $[\hat{\mathbf{x}}_1, \hat{\mathbf{x}}_2, \ldots, \hat{\mathbf{x}}_K]$ follows the distribution $\hat{p}_*$, which satisfies*

$$\mathrm{KL}\left(p_* \big\| \hat{p}_*\right) \lesssim 2e^{-2T} L \cdot (m_0 + d) + (L^2 R \eta^2 + T\eta) \cdot d + \eta m_0 + \eta K R \cdot \epsilon_{\text{score}}^2.$$

**Remark.** To achieve the KL convergence, e.g., $\mathrm{KL}\left(p_* \big\| \hat{p}_*\right) \leq \epsilon^2$ for the generated data, we require the hyper-parameters to satisfy $T = \tilde{\Theta}(1)$,

$$\eta = \tilde{\Theta}(L^{-2} d^{-1} \epsilon^{-2}) \quad \text{and} \quad \epsilon_{\text{score}} = \tilde{O}(K^{-1/2}\epsilon).$$

Under these conditions on the learning rate and score estimation error, the total gradient complexity of the inference process will be at an $\tilde{O}(KL^2 d\epsilon^{-2})$ level. Compared with typical DDPM (corresponding to the special case $K = 1$ in our setting), this complexity will have an additional factor of $K$, which means AR diffusion usually requires more inference steps to achieve the same generation quality and matches people's general perception in empirical studies.

The theorem above implies that if the score estimation error scales as $O(K^{-1/2})$, an Autoregressive (AR) diffusion model with a large $K$ cannot outperform the standard ($K = 1$) model. However, in scenarios where image patches exhibit strong correlations, partitioning the image and training multiple conditional score functions can be significantly easier than modeling the full joint distribution. Consequently, for a well-chosen patch decomposition and a sufficiently large $K$, the training loss of the AR diffusion model may satisfy $\epsilon_{\text{score}}(K) \ll K^{1/2}\epsilon_{\text{score}}(1)$, allowing it to surpass the performance of the standard model.

## 4.2 AR DIFFUSION CAPTURES CONDITIONAL DEPENDENCE

We first provide a lemma on the convergence of the AR diffusion-generated conditional distributions towards the ground truth data distribution. We then provide another lemma to contrast it against the distributions generated by vanilla diffusion models.

**Lemma 4.** *For any $k \geq 1$, for any $k$-tuples $\boldsymbol{x}_{[1:k]} \in \mathbb{R}^{d_1 + d_2 + \ldots + d_k}$, we consider the SDE. 10 to simulate the reverse process of SDE. 3, with a proper design of the time sequence $\{t_r\}_{r=0}^R$, we have*

$$\mathrm{KL}\left(p_{*,k+1|[1:k]}(\cdot|\boldsymbol{x}_{[1:k]}) \big\| \hat{p}_{*,k+1|[1:k]}(\cdot|\boldsymbol{x}_{[1:k]})\right)$$

$$\lesssim e^{-2T} \cdot \left(2L d_{k+1} + \mathbb{E}_{p_{*,k+1|[1:k]}(\cdot|\boldsymbol{x}_{[1:k]})}\left[\|\mathbf{y}\|^2\right]\right) + \eta \cdot \sum_{r=0}^{R-1} L_{k+1,r}^{\text{SM}}(\boldsymbol{\theta}|\boldsymbol{x}_{[1:k]}) + d_{k+1} L^2 R \eta^2$$

$$+ d_{k+1} T \eta + \eta \mathbb{E}_{p_{*,k+1|[1:k]}(\cdot|\boldsymbol{x}_{[1:k]})}\left[\|\mathbf{y}\|^2\right].$$

$$(15)$$

The above means given the data patch $\boldsymbol{x}_{[1:k]}$ that is being conditioned upon, we can always choose a small enough step size $\eta$ and a large enough convergence time $T$, so that the conditional distribution converges to any desired accuracy $\epsilon$: $\mathrm{KL}\left(p_{*,k+1|[1:k]}(\cdot|\boldsymbol{x}_{[1:k]}) \big\| \hat{p}_{*,k+1|[1:k]}(\cdot|\boldsymbol{x}_{[1:k]})\right) \leq \epsilon$.

On the other hand, if one only performs score matching over the joint distributions $q_{T-t}(\boldsymbol{x}_{[1:K]})$, the convergence of the diffusion model is in terms of $\mathrm{KL}\left(p_*(\boldsymbol{x}_{[1:K]}) \big\| \hat{p}_*(\boldsymbol{x}_{[1:K]})\right)$, while the conditional distributions can remain arbitrarily large.

**Lemma 5.** *Consider random vectors* $\mathbf{x}_{k+1} \in \mathbb{R}^{d_{k+1}}$ *and* $\boldsymbol{x}_{1:k} \in \mathbb{R}^{d_1+d_2+\cdots+d_k}$. *For any error threshold* $\varepsilon \in (0, 1/2]$ *and for any* $M \in \mathbb{R}$, *there exists a pair of Gaussian probability densities* $p_*, \hat{p}_* : \mathbb{R}^{d_1+d_2+\cdots+d_k} \to \mathbb{R}_+$, *such that* $\mathrm{KL}\left(p_* \middle\| \hat{p}_*\right) \le \varepsilon$, *while*

$$\mathrm{KL}\left(p_{*,k+1|[1:k]}(\cdot|\boldsymbol{x}_{[1:k]}) \middle\| \hat{p}_*(\cdot|\boldsymbol{x}_{[1:k]})\right) > M^2 \cdot \|\boldsymbol{x}_{1:k}\|^2$$

Detailed proof of Lemma 5 is provided in Appendix C. By constructing a special case where both the target and sampling distributions are Gaussian, Lemma 5 demonstrates that for vanilla diffusion models, which aim to learn the joint distribution, even when the KL divergence between the target and sampling joint distributions is constrained, the KL divergence between the corresponding conditional distributions remains lower-bounded by a constant-level value.

## 5 EXPERIMENT

Based on Section 4.2, the conditional KL in AR diffusion remains bounded (Lemma 4), whereas in diffusion it can diverge even when the joint KL between generated and data distributions are arbitrarily close (Lemma 5). Consequently, with an appropriate patch partition, AR diffusion guarantees a smaller conditional KL than non-AR diffusion, demonstrating its superior ability to capture conditional data structures. To support this claim, *we design two experiments on synthetic datasets where data dependencies are clearly defined. First*, we demonstrate a carefully designed patch partition enables AR diffusion to achieve a lower conditional KL than non-AR diffusion. This observation underscores the importance of appropriate patch partitioning, consistent with Lemma 4. Specially, the conditional KL upper bound in (15) depends on both the second moment of $p_{*,k+1|[1:k]}(\cdot|\boldsymbol{x}_{[1:k]})$ and the score-matching loss $L_{k+1,r}^{\mathrm{SM}}(\boldsymbol{\theta} \mid \boldsymbol{x}_{[1:k]})$, both explicitly tied to the patch partition $\boldsymbol{x}_{[1:k]} \in \mathbb{R}^{d_1+\cdots+d_k}$. *Second*, we compare AR and non-AR diffusion under varying patch partitions, showing how partitioning choices affect model performance.

**Conditional KL divergence experimental settings.** We conduct our experiments on MNIST by constructing training samples in the following manner. First, we concatenate four MNIST digits to form $2 \times 2$ image grids, ensuring they follow one of three arithmetic sequences, including $[0, 1, 2, 3]$, $[0, 2, 4, 6]$, or $[0, 3, 6, 9]$ in Figure 1(a). We label patches in raster-scan order (see Figure 1(b)). In all samples, *patch 0* consistently contains the same MNIST digit "0", while non-zero digits (e.g., "2") are drawn from different MNIST instances of that digit. We synthesize 3000 training samples (1:1:1 ratio across three arithmetic sequences) using 32×32-pixel images. For AR diffusion, we use a patch size of 16 with the same raster-scan order as in training. For both models, we generate 1000 samples and verify whether they form the predefined arithmetic sequence using a pre-trained MNIST classifier. More details are in Appendix E.1.

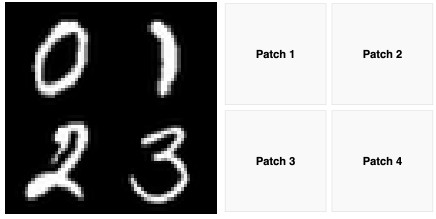

(a) Training Data     (b) Patch Index

Figure 1: **Experimental settings of the conditional KL comparison experiment.** Figure 1(a) shows the arithmetic sequence setting based on MNIST. Figure 1(b) shows the patch index of the training data.

**Results.** Table 1 shows the arithmetic sequence ratios generated by both models. Given the known uniform target distribution for the specific "0" digit, the discrete KL divergence yields a conditional KL of 0.163 for AR diffusion versus 0.890 for non-AR diffusion. These results align with our theoretical analysis: with appropriate patch partitioning, AR diffusion achieves a smaller conditional KL divergence due to its bounded conditional KL (Lemma 4), unlike unbounded conventional diffusion (Lemma 5).

| Sequences | $\mathrm{Pr}_{*,\mathrm{AR}}(\cdot|\text{'}0\text{'})$ | $\mathrm{Pr}_{*,\mathrm{non\text{-}AR}}(\cdot|\text{'}0\text{'})$ |
|---|---|---|
| $[0, 1, 2, 3]$ | 0.387 | 0.218 |
| $[0, 2, 4, 6]$ | 0.260 | 0.111 |
| $[0, 3, 6, 9]$ | 0.226 | 0.106 |
| Total | 0.873 | 0.435 |

Table 1: Conditional KL Comparison Results

**Conditional data capturing experimental settings.** In the second experiment, we design two geometric tasks: (1) **Task 1:** A square (side length $l_1$) in the upper half and a rectangle (side length $l_2$) in the lower half, with the constraint $l_2 = 1.5 l_1$; (2) **Task 2:** Two rectangles with side lengths $l_1$ (upper) and $l_2$ (lower), satisfying $l_2 = 5 l_1$. In Task 1 with $32 \times 32$ images, Figure 3(a) uses patch

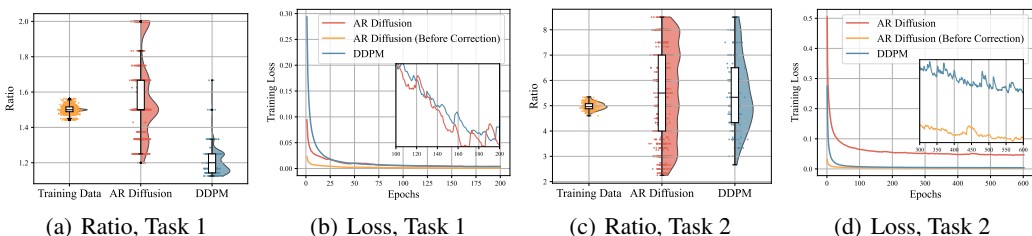

(a) Ratio, Task 1  (b) Loss, Task 1  (c) Ratio, Task 2  (d) Loss, Task 2

Figure 2: Comparison between AR Diffusion with different patch sizes and DDPM shows patch partitioning preserving data dependencies (as in Task 1) encourages AR Diffusion to generate samples closely aligned with target values (e.g., ratio = 1.5 in Task 1) and yields lower training losses.

size 16, placing the square in patch 1 and rectangle in patch 4. This spatial separation creates explicit patch dependence for the constraint $l_2 = 1.5 l_1$. In contrast, Figure 3(b) uses patch size 8, where both shapes may occupy the same patch (e.g., patch 9), making the dependence implicit and harder to learn. For each task, we train AR Diffusion and DDPM on 2000 synthetic $32 \times 32$ pixel-space samples. Object masks are extracted using pre-defiend color (e.g., red for squares, blue for rectangles in Task 1), enabling computation of side length ratios $R = l_2/l_1$ with target values $R = 1.5$ (Task 1) and $R = 5$ (Task 2). Additional implementation details are provided in Appendix E.2.

**Results.** Figures 2(a) and 2(c) show the ratio distributions and training losses for both models. In Task 1, where object separation makes feature correlations easier to learn, AR Diffusion achieves superior generation quality: its ratio distribution is sharply concentrated around the target 1.5, accompanied by lower training loss compared to DDPM. In Task 2, where object relationships are less amenable to autoregressive modeling, AR Diffusion underperforms DDPM, exhibiting a more uniform ratio distribution (vs. DDPM's sharp peak at the target 5) and higher training loss. These results confirm that when the

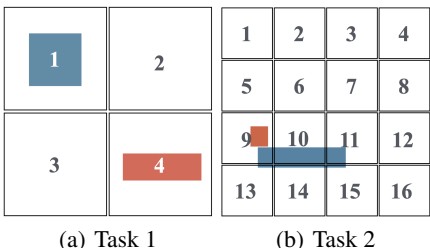

(a) Task 1   (b) Task 2

Figure 3: Various patch sizes setting.

patch partition aligns with the data structure (e.g., Task 1), AR diffusion more effectively captures conditional dependencies, yielding the smaller conditional KL established in Lemma 4. However, suboptimal patch partitioning, e.g., Task 2, increases training loss, indicating a larger $L_{k+1,r}^{SM}(\boldsymbol{\theta} \mid \boldsymbol{x}_{[1:k]})$ in the conditional KL upper bound of Lemma 4. Under these conditions, AR diffusion provides minimal or even negative gains in modeling conditional dependencies.

## 6 CONCLUSION AND LIMITATION

We present a novel theoretical and practical exploration of AR diffusion models. By formulating AR diffusion as a stage-wise auto-regressive structure, we show it can retain near-minimal assumptions on data distribution and score smoothness and converge in terms of the KL divergence for each conditional distribution, whereas vanilla diffusion models fail to preserve these conditional distributions even when the joint distribution converges.

One limitation is that we only consider the SDE-based inference, while various ODE-based inference methods have been extensively studied in typical DDPM (Li et al., 2024a; Chen et al., 2024b; Li et al., 2024b). It will be intriguing to investigate whether the Fokker-Planck equivalence can be adapted to auto-regressive settings and whether ODE-based inference can retain its convergence. Besides, the theoretical properties of some high-order ODE or SDE-based inference algorithms (Wu et al., 2024; Lu et al., 2022) are not covered in our paper, which can be left as an interesting future direction. Moreover, we directly make assumptions on the quality of the learned score function rather than proving them. Note that various works (Han et al., 2024; Chen et al., 2024a; Han et al., 2025a) have investigated the optimization, generalization properties and explore how the features are learned via denoising score matching, which can be potentially integrated with our results.

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

CONTENTS

## .1 TRAINING BEHAVIOR OF AR DIFFUSION MODEL

In order to better illustrate the difference in learning behavior of AR diffusion model, we consider a toy Gaussian model as follows:

$$\boldsymbol{x} = [\boldsymbol{x}_{[1]}, \boldsymbol{x}_{[2]}] \in \mathbb{R}^{2d}, \text{ where } \boldsymbol{x}_{[1]} \sim N(0, \mathbf{I}), \text{ and } \boldsymbol{x}_{[2]} = \boldsymbol{x}_{[1]} + \boldsymbol{\xi}, \boldsymbol{\xi} \sim N(0, \sigma^2 \mathbf{I}).$$

Clearly, we can immediately obtain that $\boldsymbol{x}$ satisfies the Gaussian distribution $\boldsymbol{x} \sim N(0, \boldsymbol{\Sigma})$, where
$$\boldsymbol{\Sigma} = \begin{bmatrix} \mathbf{I} & \mathbf{I} \\ \mathbf{I} & (1 + \sigma^2) \cdot \mathbf{I}. \end{bmatrix}$$

Then, as the optimal score function for the noisy version of $\mathbf{x}$ is a linear function, we follow the previous works (?) and consider the learnable linear function $\boldsymbol{f}(\boldsymbol{\Theta}, \boldsymbol{z}_t) = \boldsymbol{\Theta} \boldsymbol{z}_t$, where $\boldsymbol{z}_t = \beta_t \boldsymbol{x} + \sqrt{1 - \beta_t^2} \boldsymbol{\xi}$. Then, regarding the standard diffusion model, let $\{\boldsymbol{x}_1, \ldots, \boldsymbol{x}_n\}$ be the training data points, the training objective regarding the noisy data at time $t$ is given as follows:

$$L^{\text{DM}}(\boldsymbol{\Theta}) = \frac{1}{n} \sum_{i=1}^{n} \mathbb{E}_{\boldsymbol{\xi}_1, \ldots, \boldsymbol{\xi}_n} \left[ \left\| \boldsymbol{f}(\boldsymbol{\Theta}, \beta_t \boldsymbol{x}_i + [1 - \beta_t^2]^{1/2} \boldsymbol{\xi}_i) - \boldsymbol{\xi}_i \right\|_2^2 \right].$$

Taking the formula of the score function into the above loss function, we can then derive that

$$L^{\text{DM}}(\boldsymbol{\Theta}) = \frac{1}{n} \sum_{i=1}^{n} \mathbb{E}_{\boldsymbol{\xi}_1, \ldots, \boldsymbol{\xi}_n} \left[ \left\| \boldsymbol{\Theta}(\beta_t \boldsymbol{x}_i + [1 - \beta_t^2]^{1/2} \boldsymbol{\xi}_i) - \boldsymbol{\xi}_i \right\|_2^2 \right]$$

$$= \frac{1}{n} \sum_{i=1}^{n} \beta_t^2 \|\boldsymbol{\Theta} \boldsymbol{x}_i\|_2^2 + \text{tr}\left( \left([1 - \beta_t^2]^{1/2} \boldsymbol{\Theta} - \mathbf{I}\right)^\top \left([1 - \beta_t^2]^{1/2} \boldsymbol{\Theta} - \mathbf{I}\right) \right)$$

$$= \text{tr}\left( \boldsymbol{\Theta}^\top \boldsymbol{\Theta} \cdot [\beta_t^2 \hat{\boldsymbol{\Sigma}}(\boldsymbol{x}) + (1 - \beta_t^2) \mathbf{I}] \right) - 2[1 - \beta_t^2]^{1/2} \text{tr}(\boldsymbol{\Theta}) + \text{tr}(\mathbf{I}),$$

where $\hat{\boldsymbol{\Sigma}}(\boldsymbol{x}) = \frac{1}{n} \sum_{i=1}^{n} \boldsymbol{x}_i \boldsymbol{x}_i^\top$ denotes the empirical covariance matrix of the data.

Clearly, by minimizing the above loss function for diffusion model, we can obtain that the optimal model parameter $\hat{\boldsymbol{\Theta}}$ satisfies

$$\hat{\boldsymbol{\Theta}} = [1 - \beta_t^2]^{1/2} \cdot [\beta_t^2 \hat{\boldsymbol{\Sigma}}(\boldsymbol{x}) + (1 - \beta_t^2) \mathbf{I}]^{-1}.$$

Accordingly, we can obtain that the learned score function is $\boldsymbol{s}(\hat{\boldsymbol{\Theta}}, \boldsymbol{z}_t) = [1 - \beta_t^2]^{-1/2} \cdot \boldsymbol{f}(\hat{\boldsymbol{\Theta}}, \boldsymbol{z}_t) = [1 - \beta_t^2]^{-1/2} \cdot \hat{\boldsymbol{\Theta}} \boldsymbol{z}_t$.

Besides, the ground-truth score function can be also derived as $\mathbf{s}(\boldsymbol{\Theta}^*, \boldsymbol{z}_t) = [1 - \beta_t^2]^{-1/2} \cdot \boldsymbol{\Theta}^* \boldsymbol{z}_t$, where

$$\boldsymbol{\Theta}^* = [1 - \beta_t^2]^{1/2} \cdot [\beta_t^2 \boldsymbol{\Sigma}(\boldsymbol{x}) + (1 - \beta_t^2) \mathbf{I}]^{-1}.$$

Then, the corresponding score estimation error can be formulated as

$$L^{\text{SM}}(\hat{\boldsymbol{\Theta}}) = \mathbb{E}_{\boldsymbol{z}_t} \left\| \boldsymbol{s}(\hat{\boldsymbol{\Theta}}, \boldsymbol{z}_t) - \boldsymbol{s}(\boldsymbol{\Theta}^*, \boldsymbol{z}_t) \right\|_2^2$$

$$= \frac{1}{1 - \beta_t^2} \text{tr}\left[ (\hat{\boldsymbol{\Theta}} - \boldsymbol{\Theta}^*)^\top \mathbb{E}[\boldsymbol{z}_t^\top \boldsymbol{z}_t](\hat{\boldsymbol{\Theta}} - \boldsymbol{\Theta}^*) \right]$$

$$= \frac{1}{1 - \beta_t^2} \cdot \text{tr}\left[ (\hat{\boldsymbol{\Theta}} - \boldsymbol{\Theta}^*)^\top (\beta_t^2 \boldsymbol{\Sigma}(\boldsymbol{x}) + (1 - \beta_t)^2 \mathbf{I})(\hat{\boldsymbol{\Theta}} - \boldsymbol{\Theta}^*) \right]$$

$$= \text{tr}\left[ (\beta_t^2 \boldsymbol{\Sigma}(\boldsymbol{x}) + (1 - \beta_t)^2 \mathbf{I}) [\beta_t^2 \boldsymbol{\Sigma}(\boldsymbol{x}) + (1 - \beta_t^2) \mathbf{I}]^{-1} - [\beta_t^2 \hat{\boldsymbol{\Sigma}}(\boldsymbol{x}) + (1 - \beta_t^2) \mathbf{I}]^{-1} \right]^2 \right].$$

For AR diffusion model, we can set the learner functions as $\boldsymbol{s}_1(\boldsymbol{\theta}_1, \boldsymbol{z}_t) = \boldsymbol{\Theta}_1^\top \boldsymbol{z}_t$ and $\boldsymbol{s}_2(\boldsymbol{\theta}_2, \boldsymbol{z}_t, \mathbf{x}_1) = \boldsymbol{\Theta}_2^\top (\boldsymbol{z}_t - \beta_t \boldsymbol{x}_1)$. Then, the training loss functions for learning two score functions $\nabla \ln p_t(\boldsymbol{z}_t | \emptyset)$ and $\nabla \ln p_t(\boldsymbol{z}_t | \boldsymbol{x}_{[1]})$ take the form:

$$L_1^{\text{AR}}(\boldsymbol{\Theta}_1) = \frac{1}{n} \sum_{i=1}^{n} \mathbb{E}_{\boldsymbol{\xi}_1, \ldots, \boldsymbol{\xi}_n} \left[ \left\| \boldsymbol{s}_1(\boldsymbol{\Theta}_1, \beta_t \boldsymbol{x}_{i[1]} + [1 - \beta_t^2]^{1/2} \boldsymbol{\xi}_i) - \boldsymbol{\xi}_i \right\|_2^2 \right],$$

$$= \text{tr}\left( \boldsymbol{\Theta}_1^\top \boldsymbol{\Theta}_1 \cdot [\beta_t^2 \hat{\boldsymbol{\Sigma}}(\boldsymbol{x}_{[1]}) + (1 - \beta_t^2) \mathbf{I}] \right) - 2[1 - \beta_t^2]^{1/2} \text{tr}(\boldsymbol{\Theta}_1) + \text{tr}(\mathbf{I}),$$

$$L_2^{\mathrm{AR}}(\boldsymbol{\Theta}_2, \boldsymbol{x}_{[1]}) = \frac{1}{n} \sum_{i=1}^{n} \mathbb{E}_{\boldsymbol{\xi}_1, \ldots, \boldsymbol{\xi}_n} \left[ \left\| \boldsymbol{s}_2(\boldsymbol{\Theta}_2, \beta_t \boldsymbol{x}_{i[2]} + [1 - \beta_t^2]^{1/2} \boldsymbol{\xi}_i, \boldsymbol{x}_{i[1]}) - \boldsymbol{\xi}_i \right\|_2^2 \right]$$

$$= \mathrm{tr}\big(\boldsymbol{\Theta}_2^\top \boldsymbol{\Theta}_2 \cdot [\beta_t^2 \hat{\boldsymbol{\Sigma}}(\boldsymbol{x}_{[2]} - \boldsymbol{x}_{[1]}) + (1 - \beta_t^2)\mathbf{I}]\big) - 2[1 - \beta_t^2]^{1/2}\mathrm{tr}(\boldsymbol{\Theta}_2) + \mathrm{tr}(\mathbf{I}),$$

## A   NOTATIONS IN APPENDIX

**Remark 3.** *With the OU (Eq. 3) and reverse OU process (Eq. 4), standard Gaussian and* $p_{*, k+1|[1:k]}(\cdot|\boldsymbol{x}_{[1:k]})$ *can be transformed into each other. According to the closed form of Eq. 3, i.e.,*

$$\mathbf{y}_t = e^{-t} \cdot \mathbf{y}_0 + \sqrt{1 - e^{-2t}}\xi \quad \text{where} \quad \xi \sim \mathcal{N}(\mathbf{0}, \boldsymbol{I}),$$

**Training loss and conditional score estimation.**   Here, we note that the trainable parameters in the autoregressive model include

$$\boldsymbol{\theta} = [\boldsymbol{\theta}_{\mathrm{ar}}, \boldsymbol{\theta}_{\mathrm{df},1}, \ldots, \boldsymbol{\theta}_{\mathrm{df},K}].$$

To simplify the notations, we will not distinguish them strictly and only use $\boldsymbol{\theta}$ to present the trainable parameters we are concerned with. For simplicity, we take the training loss of the $k$-th random variable as an example. We first denote that

$$\boldsymbol{s}_{\boldsymbol{\theta}}(\boldsymbol{y}|t, \boldsymbol{z}) := -\sqrt{1 - e^{-2t}}\epsilon_{\boldsymbol{\theta}}(\boldsymbol{y}|t, \boldsymbol{z})$$

well-trained in the learning stage, where $\epsilon_{\boldsymbol{\theta}}$ is directly used to present the training loss provided in Li et al. (2024c), i.e.,

$$\arg \min_{\boldsymbol{\theta}} \ L(\boldsymbol{\theta})$$

$$:= \frac{1}{K} \sum_{k=1}^{K} \left[ \frac{1}{R} \sum_{r=0}^{R-1} \mathbb{E}_{\mathbf{x}_{[1:K]} \sim p_*, \xi \sim \mathcal{N}(\mathbf{0}, \boldsymbol{I}_{d_k})} \left[ \left\| \epsilon_{\boldsymbol{\theta}} \left( e^{-(T-t_r)}\mathbf{x}_k - \sqrt{1 - e^{-2(T-t_r)}}\xi \Big| T - t_r, g_{\boldsymbol{\theta}}(\mathbf{x}_{[1:k-1]}) \right) - \xi \right\|^2 \right] \right]$$

where we consider the undefined $g_{\boldsymbol{\theta}}(\mathbf{x}_{[1:0]}) = \text{None}$ to simplify the formulation. Suppose the training loss is sufficiently small for specific $\boldsymbol{\theta}_*$, i.e., $L(\boldsymbol{\theta}) \leq \epsilon_{\mathrm{score}}$, we have the following lemma

**Lemma 6.** *Under the previous notation, we have*

$$\arg \min_{\boldsymbol{\theta}} \ L(\boldsymbol{\theta}) = \arg \min_{\boldsymbol{\theta}} \ \frac{1}{K} \sum_{k=1}^{K} \mathbb{E}_{\mathbf{x}_{[1:k-1]} \sim p_{*,[1:k-1]}} \left[ \frac{1}{R} \sum_{r=0}^{R-1} (1 - e^{-2(T-t_r)}) \cdot \tilde{L}_{k,r}(\boldsymbol{\theta}|\mathbf{x}_{[1:k-1]}) \right]$$

*where*

$$\tilde{L}_{k+1,r}(\boldsymbol{\theta}|\boldsymbol{x}_{[1:k]}) = \mathbb{E}_{\mathbf{y}_0 \sim p_{*,k+1|[1:k]}(\cdot|\boldsymbol{x}_{[1:k]}), \xi \sim \mathcal{N}(\mathbf{0}, \boldsymbol{I})} \left[ \left\| \boldsymbol{s}_{\boldsymbol{\theta}}(\mathbf{y}'|(T-t_r), g_{\boldsymbol{\theta}}(\boldsymbol{x}_{[1:k]})) - \nabla \ln q_{T-t_r}(\mathbf{y}'|\boldsymbol{x}_{[1:k]}) \right\|^2 \right]$$

*and* $\mathbf{y}' = e^{-(T-t_r)} \cdot \mathbf{y}_0 + \sqrt{1 - e^{-2(T-t_t)}} \cdot \xi.$

*Proof.* For the training loss $L(\boldsymbol{\theta})$, we consider the summation component for each pair $(k, r)$, i.e.,

$$\mathbb{E}_{\mathbf{x}_{[1:K]} \sim p_*, \xi \sim \mathcal{N}(\mathbf{0}, \boldsymbol{I}_{d_k})} \left[ \left\| \xi_{\boldsymbol{\theta}} \left( e^{-(T-t_r)}\mathbf{x}_k - \sqrt{1 - e^{-2(T-t_r)}}\xi \Big| T - t_r, f_{\boldsymbol{\theta}}(\mathbf{x}_{[1:k-1]}) \right) - \xi \right\|^2 \right]$$

$$= \mathbb{E}_{\mathbf{x}_{[1:k-1]} \sim p_{*,[1:k-1]}} \left[ \mathbb{E}_{\mathbf{x}_k \sim p_{*,k|[1:k-1]}, \xi \sim \mathcal{N}(\mathbf{0}, \boldsymbol{I}_{d_k})} \left[ \left\| \xi_{\boldsymbol{\theta}} \left( e^{-(T-t_r)}\mathbf{x}_k - \sqrt{1 - e^{-2(T-t_r)}}\xi \Big| r\eta, f_{\boldsymbol{\theta}}(\mathbf{x}_{[1:k-1]}) \right) - \xi \right\|^2 \right] \right].$$

For any given $\mathbf{x}_{[1:k-1]} = \boldsymbol{x}_{[1:k-1]}$, we consider the SDE. 3, then we have

$$\mathbb{E}_{\mathbf{y}_0 \sim q_0, \xi \sim \mathcal{N}(\mathbf{0}, \boldsymbol{I}_{d_k})} \left[ \left\| \xi_{\boldsymbol{\theta}}(e^{-(T-t_r)}\mathbf{y}_0 + \sqrt{1 - e^{-2(T-t_r)}}\xi | r\eta, \boldsymbol{z}) - \xi \right\|^2 \right]$$

$$= (1 - e^{-2(T-t_r)}) \cdot \mathbb{E}_{\mathbf{y}_0, \xi} \left[ \left\| \boldsymbol{s}_{\boldsymbol{\theta}}(e^{-(T-t_r)}\mathbf{y}_0 + \sqrt{1 - e^{-2(T-t_r)}}\xi | T - t_r, \boldsymbol{z}) + \frac{\xi}{\sqrt{1 - e^{-2(T-t_r)}}} \right\|^2 \right]$$

$$= (1 - e^{-2(T-t_r)}) \cdot \mathbb{E}_{\mathbf{y}_0, \xi} \left[ \left\| \boldsymbol{s}_{\boldsymbol{\theta}}(e^{-(T-t_r)}\mathbf{y}_0 + \sqrt{1 - e^{-2(T-t_r)}}\xi | T - t_r, \boldsymbol{z}) \right\|^2 \right.$$

$$\left. + \underbrace{\frac{2}{\sqrt{1 - e^{-2(T-t_r)}}} \left\langle \boldsymbol{s}_{\boldsymbol{\theta}}(e^{-(T-t_r)}\mathbf{y}_0 + \sqrt{1 - e^{-2(T-t_r)}}\xi | T - t_r, \boldsymbol{z}), \xi \right\rangle}_{\text{Term1}} \right] + C. \tag{16}$$

Suppose $\varphi_\sigma$ to be the density function of $\mathcal{N}(\mathbf{0}, \sigma^2 \boldsymbol{I})$, considering Term 1 in Eq. 16, we have

$$\text{Term1} = \frac{2}{\sqrt{1 - e^{-2(T-t_r)}}} \cdot \int \xi \cdot \left( \int \boldsymbol{s}_{\boldsymbol{\theta}}(e^{-(T-t_r)}\boldsymbol{y}_0 + \sqrt{1 - e^{-2(T-t_r)}}\xi | (T - t_r), \boldsymbol{z}) \cdot q_0(\boldsymbol{y}_0) \mathrm{d}\boldsymbol{y}_0 \right) \cdot \varphi_1(\xi) \mathrm{d}\xi$$

$$= \frac{2}{\sqrt{1 - e^{-2(T-t_r)}}} \cdot \int q_0(\boldsymbol{y}_0) \cdot \left( \int \nabla_\xi \cdot \boldsymbol{s}_{\boldsymbol{\theta}}(e^{-(T-t_r)}\boldsymbol{y}_0 + \sqrt{1 - e^{-2(T-t_r)}}\xi | T - t_r, \boldsymbol{z}) \cdot \varphi_1(\xi) \mathrm{d}\xi \right) \mathrm{d}\boldsymbol{y}_0$$

$$= 2 \cdot \int q_0(\boldsymbol{y}_0) \cdot \left( \int \nabla_{\boldsymbol{y}'} \cdot \boldsymbol{s}_{\boldsymbol{\theta}}(\boldsymbol{y}' | T - t_r, \boldsymbol{z}) \cdot \varphi_1 \left( \frac{\boldsymbol{y}' - e^{-(T-t_r)}\boldsymbol{y}_0}{\sqrt{1 - e^{-2(T-t_r)}}} \right) \mathrm{d}\boldsymbol{y}' \right) \mathrm{d}\boldsymbol{y}_0$$

$$= 2 \cdot \int \nabla_{\boldsymbol{y}'} \cdot \boldsymbol{s}_{\boldsymbol{\theta}}(\boldsymbol{y}' | T - t_r, \boldsymbol{z}) \cdot q_{T-t_r}(\boldsymbol{y}') \mathrm{d}\boldsymbol{y}' = -2 \cdot \int \boldsymbol{s}_{\boldsymbol{\theta}}(\boldsymbol{y}' | T - t_r, \boldsymbol{z}) \cdot \nabla \ln q_{T-t_r}(\boldsymbol{y}') \cdot q_{T-t_r}(\boldsymbol{y}') \mathrm{d}\boldsymbol{y}'$$

where the first equation follows from

$$\int \langle \xi, \boldsymbol{v}(\xi) \rangle \varphi(\xi) \mathrm{d}\xi = \int \langle -\nabla \ln \varphi_1(\xi), \boldsymbol{v}(\xi) \rangle \varphi_1(\xi) \mathrm{d}\xi = -\int \langle \nabla \varphi_1(\xi), \boldsymbol{v}(\xi) \rangle \mathrm{d}\xi = \int \nabla_\xi \cdot \boldsymbol{v}(\xi) \cdot \varphi_1(\xi) \mathrm{d}\xi,$$

the second equation follows from introducing $\boldsymbol{y}' = e^{-(T-t_r)}\boldsymbol{y}_0 + \sqrt{1 - e^{-2(T-t_r)}}\xi$, and the last equation follows from integral by part. Plugging the above equation into Eq. 16, we have

$$\mathbb{E}_{\mathbf{y}_0 \sim q_0, \xi \sim \mathcal{N}(\mathbf{0}, \boldsymbol{I}_{d_k})} \left[ \left\| \epsilon_{\boldsymbol{\theta}}(e^{-(T-t_r)}\mathbf{y}_0 + \sqrt{1 - e^{-2(T-t_r)}}\xi | T - t_r, \boldsymbol{z}) - \xi \right\|^2 \right]$$

$$= (1 - e^{-2(T-t_r)}) \cdot \left( \mathbb{E}_{\mathbf{y}_0, \xi} \left[ \left\| \boldsymbol{s}_{\boldsymbol{\theta}}(e^{-(T-t_r)}\mathbf{y}_0 + \sqrt{1 - e^{-2(T-t_r)}}\xi | (T - t_r, \boldsymbol{z}) \right\|^2 \right] \right. \tag{17}$$

$$\left. - 2\mathbb{E}_{\mathbf{y}' \sim q_{(T-t_r)}} \left[ \langle \boldsymbol{s}_{\boldsymbol{\theta}}(\boldsymbol{y}' | T - t_r, \boldsymbol{z}) \cdot, \nabla \ln q_{T-t_r})(\boldsymbol{y}') \rangle \right] + C \right)$$

$$= (1 - e^{-2(T-t_r)}) \cdot \left( \mathbb{E}_{\mathbf{y}' \sim q_{T-t_r}} \left[ \| \boldsymbol{s}_{\boldsymbol{\theta}}(\mathbf{y}' | (T - t_r), \boldsymbol{z}) - \nabla \ln q_{T-t_r}(\boldsymbol{y}') \|^2 \right] + C' \right).$$

Here, we do not care about the explicit form of $C'$ since it is independent with $\boldsymbol{\theta}$ and will finally vanish with $\arg\min$ functions. According to Eq. 3, we know $\{\mathbf{y}\}_{t=0}^T$ is an OU process. Suppose

$$\tilde{L}_{k+1,r}(\boldsymbol{\theta} | \boldsymbol{x}_{[1:k]}) = \mathbb{E}_{\mathbf{y}_0 \sim p_{*,k+1|[1:k]}(\cdot | \boldsymbol{x}_{[1:k]}), \xi \sim \mathcal{N}(\mathbf{0}, \boldsymbol{I})} \left[ \left\| \boldsymbol{s}_{\boldsymbol{\theta}}(\mathbf{y}' | (T - t_r), g_{\boldsymbol{\theta}}(\boldsymbol{x}_{[1:k]})) - \nabla \ln q_{T-t_r}(\mathbf{y}' | \boldsymbol{x}_{[1:k]}) \right\|^2 \right] \tag{18}$$

and $\mathbf{y}' = e^{-(T-t_r)} \cdot \mathbf{y}_0 + \sqrt{1 - e^{-2(T-t_t)}} \cdot \xi$, then we have

$$\tilde{L}(\boldsymbol{\theta}) = \frac{1}{K} \sum_{k=1}^{K} \mathbb{E}_{\mathbf{x}_{[1:k-1]} \sim p_{*,[1:k-1]}} \left[ \frac{1}{R} \sum_{r=0}^{R-1} (1 - e^{-2(T-t_r)}) \cdot \tilde{L}_{k,r}(\boldsymbol{\theta} | \mathbf{x}_{[1:k-1]}) \right]$$

Since there is $L(\boldsymbol{\theta}) = \tilde{L}(\boldsymbol{\theta}) + C$, hence the proof is completed. $\square$

**Lemma 7.** *Suppose that we have [A2], then for any $k > 1$, $\boldsymbol{x}, \boldsymbol{x}' \in \mathbb{R}^{d_k}$ and $\boldsymbol{y} \in \mathbb{R}^{d_1 + d_2 + \ldots + d_{k-1}}$, we have*

$$\left\| \nabla \ln p_{*,k|[1:k-1]}(\boldsymbol{x} | \boldsymbol{y}) - \nabla \ln p_{*,[k|[1:k-1]]}(\boldsymbol{x}' | \boldsymbol{y}) \right\| \le 2L \cdot \| \boldsymbol{x} - \boldsymbol{x}' \|.$$

*Besides, we have $\left\| \nabla^2 \ln p_{*,[1:1]}(\cdot) \right\| \le 2L$*

*Proof.* For any $K \geq k \geq 1$, suppose the random variable

$$[\mathbf{x}_1, \mathbf{x}_2, \ldots, \mathbf{x}_k, \mathbf{x}_{k+1}, \ldots, \mathbf{x}_K] \sim p_*, \quad \mathbf{x} := [\mathbf{x}_1, \mathbf{x}_2, \ldots, \mathbf{x}_k], \quad \text{and} \quad \mathbf{y} = [\mathbf{x}_{k+1}, \ldots, \mathbf{x}_K].$$

According to the definition Eq. 1, the marginal distribution of $\mathbf{x}$ is

$$p_{*,[1:k]}(\boldsymbol{x}) = \int p_*(\boldsymbol{x}, \boldsymbol{y}) \mathrm{d}\boldsymbol{y},$$

which implies

$$\nabla_{\boldsymbol{x}} \ln p_{*,[1:k]}(\boldsymbol{x}) = \frac{\nabla_{\boldsymbol{x}} p_{*,[1:k]}(\boldsymbol{x})}{p_{*,[1:k]}(\boldsymbol{x})} = \frac{\partial_{\boldsymbol{x}} \int \exp\left(-f_*(\boldsymbol{x}, \boldsymbol{y})\right) \mathrm{d}\boldsymbol{y}}{\int \exp\left(-f_*(\boldsymbol{x}, \boldsymbol{y})\right) \mathrm{d}\boldsymbol{y}} = \frac{\int -\partial_{\boldsymbol{x}} f_*(\boldsymbol{x}, \boldsymbol{y}) \cdot \exp\left(-f_*(\boldsymbol{x}, \boldsymbol{y})\right) \mathrm{d}\boldsymbol{y}}{\int \exp\left(-f_*(\boldsymbol{x}, \boldsymbol{y})\right) \mathrm{d}\boldsymbol{y}}.$$

Furthermore, if we check its Hessian matrix, it has

$$\nabla_{\boldsymbol{x}}^2 \ln p_{*,[1:k]}(\boldsymbol{x}) = \partial_{\boldsymbol{x}} \left[ \frac{\int -\partial_{\boldsymbol{x}} f_*(\boldsymbol{x}, \boldsymbol{y}) \cdot \exp\left(-f_*(\boldsymbol{x}, \boldsymbol{y})\right) \mathrm{d}\boldsymbol{y}}{\int \exp\left(-f_*(\boldsymbol{x}, \boldsymbol{y})\right) \mathrm{d}\boldsymbol{y}} \right]$$

$$= \frac{\int \left[ -\partial_{\boldsymbol{x}}^2 f_*(\boldsymbol{x}, \boldsymbol{y}) + \partial_{\boldsymbol{x}} f_*(\boldsymbol{x}, \boldsymbol{y}) \partial_{\boldsymbol{x}}^\top f_*(\boldsymbol{x}, \boldsymbol{y}) \right] \cdot \exp\left(-f_*(\boldsymbol{x}, \boldsymbol{y})\right) \mathrm{d}\boldsymbol{y}}{\int \exp\left(-f_*(\boldsymbol{x}, \boldsymbol{y})\right) \mathrm{d}\boldsymbol{y}}$$

$$- \left[ \frac{\int \partial_{\boldsymbol{x}} f_*(\boldsymbol{x}, \boldsymbol{y}) \cdot \exp\left(-f_*(\boldsymbol{x}, \boldsymbol{y})\right) \mathrm{d}\boldsymbol{y}}{\int \exp\left(-f_*(\boldsymbol{x}, \boldsymbol{y})\right) \mathrm{d}\boldsymbol{y}} \right] \cdot \left[ \frac{\int \partial_{\boldsymbol{x}} f_*(\boldsymbol{x}, \boldsymbol{y}) \cdot \exp\left(-f_*(\boldsymbol{x}, \boldsymbol{y})\right) \mathrm{d}\boldsymbol{y}}{\int \exp\left(-f_*(\boldsymbol{x}, \boldsymbol{y})\right) \mathrm{d}\boldsymbol{y}} \right]^\top$$

$$= \mathbb{E}_{\mathbf{y} \sim p_{*,[k+1:K]|[1:k]}(\cdot|\boldsymbol{x})} \left[ -\partial_{\boldsymbol{x}}^2 f_*(\boldsymbol{x}, \mathbf{y}) \right] + \mathrm{Var}_{\mathbf{y} \sim p_{*,[k+1:K]|[1:k]}(\cdot|\boldsymbol{x})} \left[ \partial_{\boldsymbol{x}} f_*(\boldsymbol{x}, \mathbf{y}) \right].$$

Then, such a matrix can be relaxed to

$$\left\| \nabla_{\boldsymbol{x}}^2 \ln p_{*,[1:k]}(\boldsymbol{x}) \right\| \leq \left\| \mathbb{E}_{\mathbf{y} \sim p_{*,[k+1:K]|[1:k]}(\cdot|\boldsymbol{x})} \left[ \partial_{\boldsymbol{x}}^2 f_*(\boldsymbol{x}, \mathbf{y}) \right] \right\| + \left\| \mathrm{Var}_{\mathbf{y} \sim p_{*,[k+1:K]|[1:k]}(\cdot|\boldsymbol{x})} \left[ \partial_{\boldsymbol{x}} f_*(\boldsymbol{x}, \mathbf{y}) \right] \right\|$$

$$\leq \left\| \mathbb{E}_{\mathbf{y} \sim p_{*,[k+1:K]|[1:k]}(\cdot|\boldsymbol{x})} \left[ [\boldsymbol{I}, \boldsymbol{0}] \cdot \nabla^2 f_*(\boldsymbol{x}, \mathbf{y}) \cdot \begin{bmatrix} \boldsymbol{I} \\ \boldsymbol{0} \end{bmatrix} \right] \right\| + \left\| \mathbb{E}_{\mathbf{y} \sim p_{*,[k+1:K]|[1:k]}(\cdot|\boldsymbol{x})} \left[ \partial_{\boldsymbol{x}} f_*(\boldsymbol{x}, \mathbf{y}) \partial_{\boldsymbol{x}}^\top f_*(\boldsymbol{x}, \mathbf{y}) \right] \right\|$$

$$\leq \left\| \mathbb{E}_{\mathbf{y} \sim p_{*,[k+1:K]|[1:k]}(\cdot|\boldsymbol{x})} \left[ \nabla^2 f_*(\boldsymbol{x}, \mathbf{y}) \right] \right\| + \left\| \mathbb{E}_{\mathbf{y} \sim p_{*,[k+1:K]|[1:k]}(\cdot|\boldsymbol{x})} \left[ \|\partial_{\boldsymbol{x}} f_*(\boldsymbol{x}, \mathbf{y})\|^2 \cdot \boldsymbol{I} \right] \right\| \leq 2L, \tag{19}$$

where the last inequality follows from **[A2]**.

Then, suppose $k \geq 1$, we have

$$p_{*,k|[1:k-1]}(\boldsymbol{x}_k|\boldsymbol{x}_{[1:k-1]}) = \frac{p_{*,[1:k]}(\boldsymbol{x}_{[1:k]})}{\int p_{*,[1:k]}(\boldsymbol{x}_1, \boldsymbol{x}_2, \ldots, \boldsymbol{x}_k) \mathrm{d}\boldsymbol{x}_k} = p_{*,[1:k]}(\boldsymbol{x}_{[1:k]}) \cdot Z(\boldsymbol{x}_{[1:k-1]}).$$

For notation simplicity, we define $\boldsymbol{y} := (\boldsymbol{x}_1, \boldsymbol{x}_2, \ldots, \boldsymbol{x}_{k-1})$, then it has

$$\left\| \nabla \ln p_{*,k|[1:k-1]}(\boldsymbol{x}|\boldsymbol{y}) - \nabla \ln p_{*,k|[1:k-1]}(\boldsymbol{x}'|\boldsymbol{y}) \right\| = \left\| \partial_{\boldsymbol{x}} \ln p_{*,k|[1:k-1]}(\boldsymbol{y}, \boldsymbol{x}) - \partial_{\boldsymbol{x}'} \ln p_{*,k|[1:k-1]}(\boldsymbol{y}, \boldsymbol{x}') \right\|$$

$$= \left\| [\boldsymbol{0}, \boldsymbol{I}] \cdot \left[ \nabla \ln p_{*,k|[1:k-1]}(\boldsymbol{y}, \boldsymbol{x}) - \nabla \ln p_{*,k|[1:k-1]}(\boldsymbol{y}, \boldsymbol{x}') \right] \right\| \leq 2L \cdot \|\boldsymbol{x} - \boldsymbol{x}'\|$$

where the last inequality follows from the Hessian upper bound of marginal $p^{(k)}$, i.e., Eq. 19. Hence, the proof is completed. $\square$

**Lemma 8** (Bounded Second Moments). *Suppose we have [A1], then for any $K > k' \geq 1$, we have*

$$\mathbb{E}_{\mathbf{x}_{[1:k']} \sim p_{*,[1:k']}}[\|\mathbf{x}_{1:k'}\|^2] \leq m_0$$

*and*

$$\sum_{k=0}^{k'} \mathbb{E}_{\mathbf{x}_{[1:k]} \sim p_{*,[1:k]}} \left[ \mathbb{E}_{\mathbf{y} \sim p_{*,k+1|[1:k]}(\cdot|\mathbf{x}_{[1:k]})} \left[ \|\mathbf{y}\|^2 \right] \right]$$

*Proof.* For the second moment of $p_*$ and $K > k \geq 1$, we have

$$\mathbb{E}_{\mathbf{x}_{[1:k]} \sim p_{*,[1:k]}}[\|\mathbf{x}_{1:k}\|^2] = \int \|\boldsymbol{x}_{[1:k]}\|^2 \cdot p_{*,[1:k]}(\boldsymbol{x}_{[1:k]}) \mathrm{d}\boldsymbol{x}_{[1:k]}$$

$$= \int \|\boldsymbol{x}_{[1:k]}\|^2 \cdot \int p_*(\boldsymbol{x}_{[1:k]}, \boldsymbol{x}_{[k+1:K]}) \mathrm{d}\boldsymbol{x}_{[1:k]} \mathrm{d}\boldsymbol{x}_{[k+1:K]}$$

$$\leq \int \|(\boldsymbol{x}_1, \boldsymbol{x}_{[2:K-1]})\|^2 p_*(\boldsymbol{x}_{1:k}, \boldsymbol{x}_{[k+1:K]}) \mathrm{d}\boldsymbol{x}_{1:k} \mathrm{d}\boldsymbol{x}_{[k+1:K]} = m_0.$$

Besides, we have

$$\sum_{k=0}^{K-1} \mathbb{E}_{\mathbf{x}_{[1:k]} \sim p_{*,[1:k]}} \left[ \mathbb{E}_{\mathbf{y} \sim p_{*,k+1|[1:k]}(\cdot|\mathbf{x}_{[1:k]})} \left[ \|\mathbf{y}\|^2 \right] \right]$$

$$= \sum_{k=0}^{K-1} \int \|\boldsymbol{x}_{k+1}\|^2 \cdot p_{*,[1:k+1]}(\boldsymbol{x}_{[1:k]}, \boldsymbol{x}_{k+1}) \mathrm{d}(\boldsymbol{x}_{1:k}, \boldsymbol{x}_{k+1})$$

$$= \sum_{k=0}^{K-1} \int \|\boldsymbol{x}_{k+1}\|^2 p_*(\boldsymbol{x}_{[1:K]}) \mathrm{d}\boldsymbol{x}_{[1:K]} = \int \sum_{k=0}^{K-1} \|\boldsymbol{x}_{k+1}\|^2 p_*(\boldsymbol{x}_{[1:K]}) \mathrm{d}\boldsymbol{x}_{[1:K]} = m_0.$$

Hence, the proof is completed. $\qquad\qquad\square$

# B  INFERENCE COMPLEXITY

In this section, the notation of the inference process follows from those defined in Section A.

**Theorem 2.** *Suppose Assumption [A1]-[A3] hold,*

$$\delta \le \left( 0, \ln \sqrt{(4L)^{-2} + 1} + (4L)^{-1} \right]$$

*if Alg. 1 chooses the time sequence $\{\eta_r\}_{r=0}^{R-1}$ as*

$$\eta_r = \begin{cases} \eta & \textit{when} \quad 0 \le r < M \\ \eta/(1+\eta)^{r-M+1} & \textit{when} \quad M \le r < N \\ \eta & \textit{when} \quad N \le r \le R \end{cases}$$

*where*

$$M = \frac{T-1}{\eta}, \quad N = M + \frac{2\ln(1/\delta)}{\eta}, \quad \textit{and} \quad R = N + \frac{\delta}{\eta},$$

*then, the generated samples $[\hat{\mathbf{x}}_1, \hat{\mathbf{x}}_2, \ldots, \hat{\mathbf{x}}_K]$ follows the distribution $\hat{p}_*$, which satisfies*

$$\mathrm{KL}\left(p_* \big\| \hat{p}_*\right) \lesssim 2e^{-2T} L \cdot (m_0 + d)$$
$$+ (L^2 R \eta^2 + T\eta) \cdot d + \eta m_0 + \eta K R \cdot \epsilon_{\mathrm{score}}^2.$$

*Proof.* Suppose the inference process generate the sample $\hat{\mathbf{x}} = (\hat{\mathbf{x}}_1 \ldots, \hat{\mathbf{x}}_K)$ with density function $\hat{p}_*$ which satisfies

$$\hat{p}_*(\hat{\boldsymbol{x}}_{[1:K]}) = \hat{p}_{*,1}(\hat{\boldsymbol{x}}_1) \cdot \prod_{k=2}^{K} \hat{p}_{*,k|[1:k-1]}(\hat{\boldsymbol{x}}_k | \hat{\boldsymbol{x}}_{1:k-1}). \tag{20}$$

We expect to have $\mathrm{TV}\left(p_*, \hat{p}_*\right) \le \epsilon$ or $\mathrm{KL}\left(p_* \big\| \hat{p}_*\right) \le 2\epsilon^2$, which can be relaxed to

$$\mathrm{KL}\left(p_* \big\| \hat{p}_*\right) \le \mathrm{KL}\left(p_{*,[1:K-1]} \big\| \hat{p}_{*,[1:K-1]}\right)$$
$$+ \mathbb{E}_{\mathbf{x}_{[1:K-1]} \sim p_{*,[1:K-1]}} \left[ \mathrm{KL}\left(p_{*,K|[1:K-1]}(\cdot|\mathbf{x}_{[1:K-1]}) \big\| \hat{p}_{*,K|[1:K-1]}(\cdot|\mathbf{x}_{[1:K-1]}) \right) \right]$$

following from the chain rule of TV distance, i.e., Lemma 19. Within a recursive manner, we have

$$\mathrm{KL}\left(p_* \big\| \hat{p}_*\right) \le \mathrm{KL}\left(p_{*,1} \big\| \hat{p}_{*,1}\right) + \sum_{k=1}^{K-1} \mathbb{E}_{\mathbf{x}_{[1:k]} \sim p_{*,[1:k]}} \left[ \mathrm{KL}\left(p_{*,k+1|[1:k]}(\cdot|\mathbf{x}_{[1:k]}) \big\| \hat{p}_{*,k+1|[1:k]}(\cdot|\mathbf{x}_{[1:k]}) \right) \right].$$
$$\tag{21}$$

According to Lemma 9 and Remark 4, Eq. 21 can further be relaxed to

$$
\mathrm{KL}\left(p_*\|\hat{p}_*\right) \lesssim e^{-2T} \cdot \left(2Ld_1 + \mathbb{E}_{p_{*,1}}\left[\|\mathbf{y}\|^2\right]\right) + \eta \cdot \sum_{r=0}^{R-1} \tilde{L}_{1,r}(\boldsymbol{\theta}) + d_1 L^2 R \eta^2 + d_1 T\eta + \eta \mathbb{E}_{p_{*,1}}\left[\|\mathbf{y}\|^2\right]
$$

$$
+ \underbrace{\sum_{k=1}^{K-1} \mathbb{E}_{\mathbf{x}_{[1:k]}\sim p_{*,[1:k]}}\left[e^{-2T}\cdot\left(2Ld_{k+1} + \mathbb{E}_{p_{*,k+1|[1:k]}(\cdot|\boldsymbol{x}_{[1:k]})}\left[\|\mathbf{y}\|^2\right]\right)\right]}_{\text{Term 1}}
$$

$$
+ \underbrace{\sum_{k=1}^{K-1} \mathbb{E}_{\mathbf{x}_{[1:k]}\sim p_{*,[1:k]}}\left[\eta\cdot\sum_{r=0}^{R-1}\tilde{L}_{k+1,r}(\boldsymbol{\theta}|\boldsymbol{x}_{[1:k]}) + d_{k+1}L^2R\eta^2 + d_{k+1}T\eta + \eta\mathbb{E}_{p_{*,k+1|[1:k]}(\cdot|\boldsymbol{x}_{[1:k]})}\left[\|\mathbf{y}\|^2\right]\right]}_{\text{Term 2}}.
$$

$$(22)$$

For Term 1 of Eq. 22, we have

$$
\text{Term 1} = e^{-2T}\cdot\left(2L\sum_{k=1}^{K-1}d_{k+1} + \mathbb{E}_{\mathbf{x}_{[1:k]}\sim p_{*,[1:k]}}\left[\mathbb{E}_{p_{*,k+1|[1:k]}(\cdot|\boldsymbol{x}_{[1:k]})}\left[\|\mathbf{y}\|^2\right]\right]\right), \quad (23)
$$

For Term 2 of Eq. 22, with the same technique, we have

$$
\text{Term 2} = \eta\cdot\sum_{k=1}^{K-1}\sum_{r=0}^{R-1}\mathbb{E}_{\mathbf{x}_{[1:k]}\sim p_{*,[1:k]}}\left[\tilde{L}_{k+1,r}(\boldsymbol{\theta}|\boldsymbol{x}_{[1:k]})\right] + (L^2R\eta^2 + T\eta)\cdot\sum_{k=1}^{K-1}d_{k+1}
$$
$$
+ \eta\cdot\sum_{k=1}^{K}\mathbb{E}_{\mathbf{x}_{[1:k]}\sim p_{*,[1:k]}}\left[\mathbb{E}_{p_{*,k+1|[1:k]}(\cdot|\boldsymbol{x}_{[1:k]})}\left[\|\mathbf{y}\|^2\right]\right]. \quad (24)
$$

Plugging Eq. 23 and Eq. 24 into Eq. 22, we have

$$
\mathrm{KL}\left(p_*\|\hat{p}_*\right) \lesssim 2e^{-2T}L\cdot\sum_{k=0}^{K-1}\left(d_{k+1} + \mathbb{E}_{\mathbf{x}_{[1:k]}\sim p_{*,[1:k]}}\left[\mathbb{E}_{\mathbf{y}\sim p_{*,k+1|[1:k]}(\cdot|\mathbf{x}_{[1:k]})}\left[\|\mathbf{y}\|^2\right]\right]\right)
$$
$$
+ \eta\cdot\sum_{k=0}^{K-1}\sum_{r=0}^{R-1}\mathbb{E}_{\mathbf{x}_{[1:k]}\sim p_{*,[1:k]}}\left[\tilde{L}_{k+1,r}(\boldsymbol{\theta}|\boldsymbol{x}_{[1:k]})\right] + (L^2R\eta^2 + T\eta)\cdot\sum_{k=0}^{K-1}d_{k+1}
$$
$$
+ \eta\cdot\sum_{k=0}^{K}\mathbb{E}_{\mathbf{x}_{[1:k]}\sim p_{*,[1:k]}}\left[\mathbb{E}_{p_{*,k+1|[1:k]}(\cdot|\boldsymbol{x}_{[1:k]})}\left[\|\mathbf{y}\|^2\right]\right]
$$
$$
\leq 2e^{-2T}L\cdot\left(m_0 + \sum_{k=1}^{K}d_k\right) + \frac{\eta K R}{K R}\cdot\sum_{k=0}^{K-1}\sum_{r=0}^{R-1}\mathbb{E}_{\mathbf{x}_{[1:k]}\sim p_{*,[1:k]}}\left[\tilde{L}_{k+1,r}(\boldsymbol{\theta}|\boldsymbol{x}_{[1:k]})\right]
$$
$$
+ (L^2R\eta^2 + T\eta)\cdot\sum_{k=0}^{K-1}d_{k+1} + \eta m_0
$$
$$
\leq 2e^{-2T}L\cdot\left(m_0 + \sum_{k=1}^{K}d_k\right) + (L^2R\eta^2 + T\eta)\cdot\sum_{k=0}^{K-1}d_{k+1} + \eta m_0 + \eta K R\cdot\epsilon_{\text{score}}^2
$$

where the second inequality follows from Lemma 8 and the last inequality follows from a small training error, i.e., **[A3]**. Without loss of generality, we suppose $L \geq 1$,

$$
d = \sum_{k=1}^{K}d_k \quad \text{and} \quad c = \ln\frac{\sqrt{\frac{1}{4L^2}+4}+\frac{1}{2L}}{2} < 1,
$$

then by requiring

$$T = \Theta\left(\ln\frac{8L(m_0+d)}{\epsilon^2}\right), \quad R\eta = \Theta(T + \ln 1/c) = \Theta\left(\ln\frac{16L(m_0+d)}{\epsilon^2 \cdot \sqrt{\frac{1}{4L^2}+4}+\frac{1}{2L}}\right),$$

$$\epsilon_{\text{score}} = \Theta\left(\frac{\epsilon}{2\sqrt{K\cdot R\eta}}\right) = \Theta\left(\frac{\epsilon}{2\sqrt{K}}\cdot\left(\ln\frac{16L(m_0+d)}{\epsilon^2\cdot\sqrt{\frac{1}{4L^2}+4}+\frac{1}{2L}}\right)^{-1/2}\right), \quad \text{and} \quad (25)$$

$$\eta = O\left(\frac{\epsilon^2}{4L^2(R\eta)d}\right) = O\left(\frac{\epsilon^2}{4L^2d}\cdot\left(\ln\frac{16L(m_0+d)}{\epsilon^2\cdot\sqrt{\frac{1}{4L^2}+4}+\frac{1}{2L}}\right)^{-1}\right),$$

we will have $\text{KL}\left(p_*\|\hat{p}_*\right) \lesssim \epsilon^2$ and $\text{TV}\left(p_*, \hat{p}_*\right) \leq \epsilon$. To calculate the gradient complexity, which is

$$O\left(KR\right) = O\left(\frac{KR\eta}{\eta}\right) = O\left(K\cdot\ln\frac{16L(m_0+d)}{\epsilon^2\cdot\sqrt{\frac{1}{4L^2}+4}+\frac{1}{2L}}\cdot\frac{4L^2d}{\epsilon^2}\cdot\ln\frac{16L(m_0+d)}{\epsilon^2\cdot\sqrt{\frac{1}{4L^2}+4}+\frac{1}{2L}}\right)$$

$$= O\left(\frac{KL^2d}{\epsilon^2}\cdot\ln\frac{L(m_0+d)}{\epsilon}\right).$$

$$(26)$$

Hence, the proof is completed. □

**Lemma 9.** *For any $k \geq 1$, for any $k$-tuples $\boldsymbol{x}_{[1:k]} \in \mathbb{R}^{d_1+d_2+\ldots+d_k}$, we consider the SDE. 10 to simulate the reverse process of SDE. 3, then we have*

$$\text{KL}\left(p_{*,k+1|[1:k]}(\cdot|\mathbf{x}_{[1:k]})\big\|\hat{p}_{*,k+1|[1:k]}(\cdot|\mathbf{x}_{[1:k]})\right) \lesssim e^{-2T}\cdot\left(2Ld_{k+1}+\mathbb{E}_{p_{*,k+1|[1:k]}(\cdot|\boldsymbol{x}_{[1:k]})}\left[\|\mathbf{y}\|^2\right]\right)$$

$$+\eta\cdot\sum_{r=0}^{R-1}\tilde{L}_{k+1,r}(\boldsymbol{\theta}|\boldsymbol{x}_{[1:k]}) + d_{k+1}L^2R\eta^2 + d_{k+1}T\eta + \eta\mathbb{E}_{p_{*,k+1|[1:k]}(\cdot|\boldsymbol{x}_{[1:k]})}\left[\|\mathbf{y}\|^2\right].$$

*Proof.* Similar as Benton et al. (2024) and Chen et al. (2023a), we consider the step size satisfying $\eta_r \leq \eta\min(1, T - t_{r+1})$. $\eta$ is the parameter for controlling the maximum step size. We denote the conditional variance of $\mathbf{y}_t$ given $\mathbf{y}_s$ as $\sigma_{s,t}^2 := 1 - e^{-2(t-s)}$ and the conditional expectation as $\alpha_{s,t} := \text{E}(\mathbf{y}_t|\mathbf{y}_s) = e^{-(t-s)}$ where $0 \leq s \leq t \leq T$. And we denote the posterior mean as $\boldsymbol{\mu}_t := \mathbb{E}_{q_{0|t}}(\mathbf{y}_0)$ and posterior variance as $\boldsymbol{\Sigma}_t := \text{Cov}_{q_{0|t}}(\mathbf{y}_0)$. Suppose $\hat{p}_{*,k+1|[1:k]}(\cdot|\boldsymbol{x}_{[1:k]}) = \hat{q}_T(\cdot)$, we have

$$\text{KL}\left(p_{*,k+1|[1:k]}(\cdot|\mathbf{x}_{[1:k]})\big\|\hat{p}_{*,k+1|[1:k]}(\cdot|\mathbf{x}_{[1:k]})\right)$$

$$= \text{KL}\left(q_T^{\leftarrow}\big\|\hat{q}_T\right) \leq \text{KL}\left(q_{t_{R-1}}^{\leftarrow}\big\|\hat{q}_{t_{R-1}}\right) + \mathbb{E}_{\mathbf{y}^{\leftarrow}\sim q_{t_{R-1}}^{\leftarrow}}\left[\text{KL}\left(q_{t_R|t_{R-1}}^{\leftarrow}(\cdot|\mathbf{y}^{\leftarrow})\big\|\hat{q}_{t_R|t_{R-1}}(\cdot|\mathbf{y}^{\leftarrow})\right)\right]$$

$$\leq \text{KL}\left(q_0^{\leftarrow}\big\|\hat{q}_0\right) + \underbrace{\sum_{r=0}^{R-1}\mathbb{E}_{\mathbf{y}^{\leftarrow}\sim q_{t_r}^{\leftarrow}}\left[\text{KL}\left(q_{t_{r+1}|t_r}^{\leftarrow}(\cdot|\tilde{\mathbf{y}})\big\|\hat{q}_{t_{r+1}|t_r}(\cdot|\tilde{\mathbf{y}})\right)\right]}_{\text{reverse transition error}}$$

$$(27)$$

where the first inequality follows from the chain rule, i.e., Lemma 19, of KL divergence, and the second one follows from the recursive manner. Besides, we have $\hat{q}_0 = \varphi_1$ which denotes the density function of $\mathcal{N}(\mathbf{0}, \boldsymbol{I})$.

**Initialization Error.** We first consider to upper bound $\text{TV}(q_T, \varphi_1)$. Due to Lemma 10, we have

$$\text{KL}\left(q_0^{\leftarrow}\big\|\hat{q}_0\right) = \text{KL}\left(q_T\big\|\varphi_1\right) \leq e^{-2T}\cdot\text{KL}\left(p_{k+1|[1:k],0}^{\rightarrow}(\cdot|\boldsymbol{x}_{1:k})\big\|\varphi_1\right)$$

$$\leq e^{-2T}\cdot\left(2Ld_{k+1}+\mathbb{E}_{p_{*,k+1|[1:k]}(\cdot|\boldsymbol{x}_{[1:k]})}\left[\|\mathbf{y}\|^2\right]\right).$$

$$(28)$$

**Reverse Transition Error.** According to Lemma 11, the reverse transition error can be relaxed as

$$\sum_{r=0}^{R-1} \mathbb{E}_{\mathbf{y}^{\leftarrow} \sim q_{t_r}^{\leftarrow}} \left[ \mathrm{KL}\left( q_{t_{r+1}|t_r}^{\leftarrow}(\cdot|\mathbf{y}^{\leftarrow}) \big\| \hat{q}_{t_{r+1}|t_r}(\cdot|\mathbf{y}^{\leftarrow}) \right) \right]$$

$$\leq \sum_{r=0}^{R-1} \mathbb{E}_{\mathbf{y}_{[0:T]}^{\leftarrow} \sim Q_{[0:T]}^{\leftarrow}} \left[ \int_{t_r}^{t_{r+1}} \left\| s_{\boldsymbol{\theta}}(\mathbf{y}_{t_r}^{\leftarrow}|T-t_r,\boldsymbol{z}) - \nabla \ln q_{T-t}(\mathbf{y}_t^{\leftarrow}) \right\|^2 \mathrm{d}t \right]$$

$$\leq \underbrace{\sum_{r=0}^{R-1} \eta_r \cdot \mathbb{E}_{\mathbf{y}_{t_r}^{\leftarrow} \sim q_{t_r}^{\leftarrow}} \left[ \left\| s_{\boldsymbol{\theta}}(\mathbf{y}_{t_r}^{\leftarrow}|T-t_r,\boldsymbol{z}) - \nabla \ln q_{T-t_r}(\mathbf{y}_{t_r}^{\leftarrow}) \right\|^2 \right]}_{\text{score estimation error}} \tag{29}$$

$$+ \underbrace{\sum_{r=0}^{R-1} \mathbb{E}_{\tilde{\mathbf{y}}_{[0:T]} \sim \tilde{Q}_{[0:T]}} \left[ \int_{t_r}^{t_{r+1}} \left\| \nabla \ln q_{T-t_r}(\tilde{\mathbf{y}}_{t_r}) - \nabla \ln q_{T-t}(\tilde{\mathbf{y}}_t) \right\|^2 \mathrm{d}t \right]}_{\text{discretization error}}$$

Since we have $q_{T-t}^{\leftarrow} = q_t$ with Eq. 4, the score estimation error satisfies

$$\sum_{r=0}^{R-1} \eta_r \cdot \mathbb{E}_{\mathbf{y}_{t_r}^{\leftarrow} \sim q_{t_r}^{\leftarrow}} \left[ \left\| s_{\boldsymbol{\theta}}(\mathbf{y}_{t_r}^{\leftarrow}|T-t_r,\boldsymbol{z}) - \nabla \ln q_{T-t_r}(\mathbf{y}_{t_r}^{\leftarrow}) \right\|^2 \right]$$

$$\leq \eta \cdot \sum_{r=0}^{R-1} \mathbb{E}_{\mathbf{y}_{t_r}^{\leftarrow} \sim q_{T-t_r}} \left[ \left\| s_{\boldsymbol{\theta}}(\mathbf{y}_{t_r}^{\leftarrow}|T-t_r,\boldsymbol{z}) - \nabla \ln q_{T-t_r}(\mathbf{y}_{t_r}^{\leftarrow}) \right\|^2 \right]$$

$$= \eta \cdot \sum_{r=0}^{R-1} \mathbb{E}_{\mathbf{x}' \sim p_{k+1|[1:k],t_r}^{\rightarrow}(\cdot|\boldsymbol{x}_{[1:k]})} \left[ \left\| s_{\boldsymbol{\theta}}(\mathbf{x}'|T-t_r, g_{\boldsymbol{\theta}}(\boldsymbol{x}_{[1:k]})) - \nabla \ln p_{k+1|[1:k],T-t_r}^{\rightarrow}(\mathbf{x}'|\boldsymbol{x}_{[1:k]}) \right\|^2 \right]$$

$$= \eta \cdot \sum_{r=0}^{R-1} \tilde{L}_{k+1,r}(\boldsymbol{\theta}|\boldsymbol{x}_{[1:k]}) \tag{30}$$

where the second inequality follows from the definition $q_t = q_{T-t}^{\leftarrow}$ and the last equation follows from the definition Eq. 18 in Lemma 6. Considering the discretization error, we have

$$\sum_{r=0}^{R-1} \mathbb{E}_{\mathbf{y}_{[0:T]}^{\leftarrow} \sim Q_{[0:T]}^{\leftarrow}} \left[ \int_{t_r}^{t_{r+1}} \left\| \nabla \ln q_{T-t_r}(\mathbf{y}_{t_r}^{\leftarrow}) - \nabla \ln q_{T-t}(\mathbf{y}_t^{\leftarrow}) \right\|^2 \mathrm{d}t \right] \tag{31}$$

$$\lesssim d_{k+1} L^2 R \eta^2 + d_{k+1} T \eta + \eta \mathbb{E}_{p_{*,k+1|[1:k]}(\cdot|\boldsymbol{x}_{[1:k]})} \left[ \|\mathbf{y}\|^2 \right]$$

due to Lemma 12. Therefore, plugging Eq. 30 and Eq. 31 into Eq. 29, we have

$$\sum_{r=0}^{R-1} \mathbb{E}_{\mathbf{y}^{\leftarrow} \sim q_{t_r}^{\leftarrow}} \left[ \mathrm{KL}\left( q_{t_{r+1}|t_r}^{\leftarrow}(\cdot|\mathbf{y}^{\leftarrow}) \big\| \hat{q}_{t_{r+1}|t_r}(\cdot|\mathbf{y}^{\leftarrow}) \right) \right] \tag{32}$$

$$\lesssim \eta \cdot \sum_{r=0}^{R-1} \tilde{L}_{k+1,r}(\boldsymbol{\theta}|\boldsymbol{x}_{[1:k]}) + d_{k+1} L^2 R \eta^2 + d_{k+1} T \eta + \eta \mathbb{E}_{p_{*,k+1|[1:k]}(\cdot|\boldsymbol{x}_{[1:k]})} \left[ \|\mathbf{y}\|^2 \right].$$

Combining with Eq. 28, Eq. 27 can be written as

$$\mathrm{KL}\left( p_{*,k+1|[1:k]}(\cdot|\mathbf{x}_{[1:k]}) \big\| \hat{p}_{*,k+1|[1:k]}(\cdot|\mathbf{x}_{[1:k]}) \right) \lesssim e^{-2T} \cdot \left( 2L d_{k+1} + \mathbb{E}_{p_{*,k+1|[1:k]}(\cdot|\boldsymbol{x}_{[1:k]})} \left[ \|\mathbf{y}\|^2 \right] \right)$$

$$+ \eta \cdot \sum_{r=0}^{R-1} \tilde{L}_{k+1,r}(\boldsymbol{\theta}|\boldsymbol{x}_{[1:k]}) + d_{k+1} L^2 R \eta^2 + d_{k+1} T \eta + \eta \mathbb{E}_{p_{*,k+1|[1:k]}(\cdot|\boldsymbol{x}_{[1:k]})} \left[ \|\mathbf{y}\|^2 \right].$$

and the proof is completed. $\square$

**Remark 4.** *In Lemma 9, we require $k \geq 1$ and calculate the upper bound between the conditional transition kernel when $\mathbf{x}_{[1:k]} = \boldsymbol{x}_{[1:k]}$. While this lemma can easily be adapted to the case for the unconditional generation of the distribution $\hat{p}_{*,1}$. In the process to generate $\hat{\mathbf{x}}_1$, we should only*

consider Eq. 3 – Eq. 10 as $q_0 = q_T^\leftarrow = p_{*,1}$ rather than $q_0 = \hat{q}_T = p_{*,k+1|[1:k]}(\cdot|\boldsymbol{x}_{[1:k]})$. Then, suppose $\hat{p}_{*,1}(\boldsymbol{x}) = \hat{q}_T(\boldsymbol{x})$, we have

$$
\mathrm{KL}\left(p_{*,1}\big\|\hat{p}_{*,1}\right) = \mathrm{KL}\left(q_T^\leftarrow\big\|\hat{q}_T\right) \leq \mathrm{KL}\left(q_{t_{R-1}}^\leftarrow\big\|\hat{q}_{t_{R-1}}\right) + \mathbb{E}_{\mathbf{y}^\leftarrow \sim q_{t_{R-1}}^\leftarrow}\left[\mathrm{KL}\left(q_{t_R|t_{R-1}}^\leftarrow(\cdot|\mathbf{y}^\leftarrow)\big\|\hat{q}_{t_R|t_{R-1}}(\cdot|\mathbf{y}^\leftarrow)\right)\right]
$$

$$
\leq \mathrm{KL}\left(q_0^\leftarrow\big\|\hat{q}_0\right) + \underbrace{\sum_{r=0}^{R-1}\mathbb{E}_{\mathbf{y}^\leftarrow \sim q_{t_r}^\leftarrow}\left[\mathrm{KL}\left(q_{t_{r+1}|t_r}^\leftarrow(\cdot|\mathbf{y}^\leftarrow)\big\|\hat{q}_{t_{r+1}|t_r}(\cdot|\mathbf{y}^\leftarrow)\right)\right]}_{\text{reverse transition error}}
$$

due to Lemma 20. The control of $\mathrm{KL}\left(q_0^\leftarrow\big\|\hat{q}_0\right)$ is similar to Eq. 28, and have

$$
\mathrm{KL}\left(q_0^\leftarrow\big\|\hat{q}_0\right) = \mathrm{KL}\left(q_T\big\|\varphi_1\right) \leq e^{-2T} \cdot \mathrm{KL}\left(p_{k+1|[1:k],0}^\rightarrow(\cdot|\boldsymbol{x}_{1:k})\big\|\varphi_1\right) \leq e^{-2T} \cdot \left(2Ld_1 + \mathbb{E}_{p_{*,1}}\left[\|\mathbf{y}\|^2\right]\right)
$$

where the last inequality follows from Lemma 8. Additionally, the control of the expected conditional KL divergence gap is similar to Eq. 32, which satisfies

$$
\sum_{r=0}^{R-1}\mathbb{E}_{\mathbf{y}^\leftarrow \sim q_{t_r}^\leftarrow}\left[\mathrm{KL}\left(q_{t_{r+1}|t_r}^\leftarrow(\cdot|\mathbf{y}^\leftarrow)\big\|\hat{q}_{t_{r+1}|t_r}(\cdot|\mathbf{y}^\leftarrow)\right)\right] \lesssim \eta \cdot \sum_{r=0}^{R-1}\tilde{L}_{1,r}(\boldsymbol{\theta}) + d_1 L^2 R\eta^2 + d_1 T\eta + \eta\mathbb{E}_{p_{*,1}}\left[\|\mathbf{y}\|^2\right].
$$

Here, with a little abuse of notation, we have

$$
\tilde{L}_{1,r}(\boldsymbol{\theta}) = \tilde{L}_{1,r}(\boldsymbol{\theta}|\boldsymbol{x}_{[1:0]}) \quad \text{and} \quad \boldsymbol{x}_{[1:0]} = \text{none}
$$

in the training loss. Therefore, in summary, we have

$$
\mathrm{KL}\left(p_{*,1}\big\|\hat{p}_{*,1}\right) \leq e^{-2T} \cdot \left(2Ld_1 + \mathbb{E}_{p_{*,1}}\left[\|\mathbf{y}\|^2\right]\right) + \eta \cdot \sum_{r=0}^{R-1}\tilde{L}_{1,r}(\boldsymbol{\theta}) + d_1 L^2 R\eta^2 + d_1 T\eta + \eta\mathbb{E}_{p_{*,1}}\left[\|\mathbf{y}\|^2\right].
$$
(33)

**Lemma 10** (Adapted from Theorem 4 in Vempala & Wibisono (2019)). *Along the Langevin dynamics SDE. 3, for any $t \in [0,T]$ and $k \geq 1$, we have*

$$
\mathrm{KL}\left(p_{k+1|[1:k],t}^\rightarrow(\cdot|\boldsymbol{x}_{1:k})\big\|\varphi_1\right) \leq e^{-2t} \cdot \left(2Ld + \mathbb{E}_{p_{*,k+1|[1:k]}(\cdot|\boldsymbol{x}_{1:k})}\left[\|\mathbf{y}\|^2\right]\right)
$$

*Proof.* The Fokker-Planck equation of SDE. 3, i.e.,

$$
\frac{\partial p_{k+1|[1:k],t}^\rightarrow(\boldsymbol{y}|\boldsymbol{x}_{1:k})}{\partial t} = \nabla \cdot \left(p_{k+1|[1:k],t}^\rightarrow(\boldsymbol{y}|\boldsymbol{x}_{1:k}) \cdot \boldsymbol{y}\right) + \Delta p_{k+1|[1:k],t}^\rightarrow(\boldsymbol{y}|\boldsymbol{x}_{1:k})
$$

$$
= \nabla \cdot \left(p_{k+1|[1:k],t}^\rightarrow(\boldsymbol{y}|\boldsymbol{x}_{1:k})\nabla \ln \frac{p_{k+1|[1:k],t}^\rightarrow(\boldsymbol{y}|\boldsymbol{x}_{1:k})}{e^{-\|\boldsymbol{y}\|^2/2}}\right)
$$

denotes its stationary distribution follows from the standard Gaussian $\mathcal{N}(\mathbf{0}, \boldsymbol{I})$ with density function $\varphi_1$. Due to Lemma 17, the 1-strongly log-concave standard Gaussian satisfies LSI with a constant 1, which means for any distribution with density function $p$, we have

$$
\mathrm{KL}\left(q\big\|\varphi_1\right) \leq \mathrm{FI}\left(q\big\|\varphi_1\right) = \frac{1}{2}\int q(\boldsymbol{x})\left\|\nabla \ln \frac{q(\boldsymbol{y})}{\varphi_1(\boldsymbol{y})}\right\|^2 \mathrm{d}\boldsymbol{x},
$$

which implies

$$
\frac{\mathrm{dKL}\left(p_{k+1|[1:k],t}^\rightarrow(\boldsymbol{y}|\boldsymbol{x}_{1:k})\big\|\varphi_1\right)}{\mathrm{d}t} = -\int p_{k+1|[1:k],t}^\rightarrow(\boldsymbol{y}|\boldsymbol{x}_{1:k})\left\|\nabla \ln \frac{p_{k+1|[1:k],t}^\rightarrow(\boldsymbol{y}|\boldsymbol{x}_{1:k})}{\varphi_1(\boldsymbol{y})}\right\|^2 \mathrm{d}\boldsymbol{y}
$$

$$
\leq 2\mathrm{KL}\left(p_{k+1|[1:k],t}^\rightarrow(\boldsymbol{y}|\boldsymbol{x}_{1:k})\big\|\varphi_1\right).
$$

According to Grönwall's inequality, it has

$$
\mathrm{KL}\left(p_{k+1|[1:k],t}^\rightarrow(\cdot|\boldsymbol{x}_{1:k})\big\|\varphi_1\right) \leq e^{-2t} \cdot \mathrm{KL}\left(p_{k+1|[1:k],0}^\rightarrow(\cdot|\boldsymbol{x}_{1:k})\big\|\varphi_1\right)
$$
(34)

where the RHS satisfies

$$\mathrm{KL}\left(p^{\rightarrow}_{k+1|[1:k],0}(\cdot|\boldsymbol{x}_{1:k})\big\|\varphi_1\right) = \mathrm{KL}\left(p_{*,k+1|[1:k]}(\cdot|\boldsymbol{x}_{[1:k]})\big\|\varphi_1\right) \le \mathrm{FI}\left(p_{*,k+1|[1:k]}(\cdot|\boldsymbol{x}_{[1:k]})\big\|\varphi_1\right)$$

$$= \int p_{*,k+1|[1:k]}(\boldsymbol{y}|\boldsymbol{x}_{[1:k]}) \cdot \left\|\nabla \ln \frac{p_{*,k|[1:k]}(\boldsymbol{y}|\boldsymbol{x}_{[1:k]})}{\exp(-\|\boldsymbol{y}\|^2/2)}\right\| \mathrm{d}\boldsymbol{y}$$

$$\le 2Ld + \mathbb{E}_{p_{*,k+1|[1:k]}(\cdot|\boldsymbol{x}_{[1:k]})}\left[\|\mathbf{y}\|^2\right].$$

$$(35)$$

The last inequality follows from the combination of Lemma 7 and Lemma 18. Then, combining Eq. 34 and Eq. 35, for any $t \in [0, T]$, we have

$$\mathrm{KL}\left(p^{\rightarrow}_{k+1|[1:k],t}(\cdot|\boldsymbol{x}_{1:k})\big\|\varphi_1\right) \le e^{-2t} \cdot \left(2Ld + \mathbb{E}_{p_{*,k+1|[1:k]}(\cdot|\boldsymbol{x}_{[1:k]})}\left[\|\mathbf{y}\|^2\right]\right)$$

and the proof is completed. □

**Lemma 11.** *With the same notation in Lemma B.2, we have*

$$\mathbb{E}_{\mathbf{y}\sim\tilde{q}_{t_r}}\left[\mathrm{KL}\left(q^{\leftarrow}_{t_{r+1}|t_r}(\cdot|\mathbf{y})\big\|\hat{q}_{t_{r+1}|t_r}(\cdot|\mathbf{y})\right)\right] \le \frac{1}{2}\int_{t_r}^{t_{r+1}}\mathbb{E}_{(\mathbf{y},\mathbf{y}')\sim q^{\leftarrow}_{t_r,t}}\left[\|\nabla\ln q^{\leftarrow}_t(\mathbf{y}') - \boldsymbol{s}_{\boldsymbol{\theta}}(\mathbf{y}|T - t_r, \boldsymbol{z})\|^2\right]\mathrm{d}t$$

*Proof.* Let's consider the process 4 and 10, by Lemma 21 and 22, for $t \in (t_r, t_{r+1}]$ and $\mathbf{y}^{\leftarrow}_{t_r} = \boldsymbol{y}$, we have

$$\frac{\partial}{\partial t}\mathrm{KL}\left(q^{\leftarrow}_{t|t_r}(\cdot|\boldsymbol{y})\big\|\hat{q}_{t|t_r}(\cdot|\boldsymbol{y})\right) \tag{36}$$

$$= -\mathbb{E}_{\mathbf{y}'\sim q^{\leftarrow}_{t|t_r}(\cdot|\boldsymbol{y})}\left\|\nabla\ln\frac{q^{\leftarrow}_t(\mathbf{y}'|\boldsymbol{y})}{\hat{q}_t(\mathbf{y}'|\boldsymbol{y})}\right\|^2 + \mathbb{E}_{\mathbf{y}'\sim q^{\leftarrow}_{t|t_r}(\cdot|\boldsymbol{y})}\left[\left\langle\nabla\ln q^{\leftarrow}_t(\mathbf{y}') - \boldsymbol{s}_{\boldsymbol{\theta}}(\boldsymbol{y}|T - t_r, \boldsymbol{z}), \nabla\ln\frac{q^{\leftarrow}_{t|t_r}(\mathbf{y}'|\boldsymbol{y})}{\hat{q}_{t|t_r}(\mathbf{y}'|\boldsymbol{y})}\right\rangle\right]$$

$$\le \frac{1}{2}\mathbb{E}_{\mathbf{y}'\sim q^{\leftarrow}_{t|t_r}(\cdot|\boldsymbol{y})}\left[\|\nabla\ln q^{\leftarrow}_t(\mathbf{y}') - \boldsymbol{s}_{\boldsymbol{\theta}}(\boldsymbol{y}|T - t_r, \boldsymbol{z})\|^2\right]. \tag{37}$$

The last inequality holds by the fact that $\langle\mathbf{w}, \mathbf{v}\rangle \le \frac{1}{2}\|\mathbf{w}\|^2 + \frac{1}{2}\|\mathbf{v}\|^2$. Integrating both sides of Eq. 36 and utilizing Lemma 22, we obtain

$$\mathrm{KL}\left(q^{\leftarrow}_{t_{r+1}|t_r}(\cdot|\boldsymbol{y})\big\|\hat{q}_{t_{r+1}|t_r}(\cdot|\boldsymbol{y})\right) \le \frac{1}{2}\int_{t_r}^{t_{r+1}}\mathbb{E}_{\mathbf{y}'\sim q^{\leftarrow}_{t|t_r}(\cdot|\boldsymbol{y})}\left[\|\nabla\ln q^{\leftarrow}_t(\mathbf{y}') - \boldsymbol{s}_{\boldsymbol{\theta}}(\boldsymbol{y}|T - t_r, \boldsymbol{z})\|^2\right]\mathrm{d}t.$$

Then, integrating on the both sides w.r.t. $\tilde{q}_{t_r}$ yields

$$\mathbb{E}_{\mathbf{y}\sim q^{\leftarrow}_{t_r}}\left[\mathrm{KL}\left(q^{\leftarrow}_{t_{r+1}|t_r}(\cdot|\mathbf{y})\big\|\hat{q}_{t_{r+1}|t_r}(\cdot|\mathbf{y})\right)\right]$$

$$\le \frac{1}{2}\int_{t_r}^{t_{r+1}}\mathbb{E}_{(\mathbf{y},\mathbf{y}')\sim q^{\leftarrow}_{t_r,t}}\left[\|\nabla\ln q^{\leftarrow}_t(\mathbf{y}') - \boldsymbol{s}_{\boldsymbol{\theta}}(\mathbf{y}|T - t_r, \boldsymbol{z})\|^2\right]\mathrm{d}t.$$

$$(38)$$

□

**Lemma 12.** *Assume $L \ge 1$, for step sizesatisfies*

$$\eta_r \le \min\{1, \eta, \eta(T - t_{r+1})\},$$

*the discretization error in Eq. 29 can be bounded as*

$$\sum_{r=0}^{R-1}\mathbb{E}_{\mathbf{y}^{\leftarrow}_{[0:T]}\sim Q^{\leftarrow}_{[0:T]}}\left[\int_{t_r}^{t_{r+1}}\left\|\nabla\ln q_{T-t_r}(\mathbf{y}^{\leftarrow}_{t_r}) - \nabla\ln q_{T-t}(\mathbf{y}^{\leftarrow}_t)\right\|^2\mathrm{d}t\right]$$

$$\lesssim dL^2R\eta^2 + dT\eta + \mathbb{E}_{p_{*,k+1|[1:k]}(\cdot|\boldsymbol{x}_{[1:k]})}\left[\|\mathbf{y}\|^2\right]\eta.$$

*Proof.* Following Benton et al. (2024) and Chen et al. (2023a), we divide our time period into three intervals: $[0, T - 1]$, $(T - 1, T - \delta]$, and $(T - \delta, T]$, and treat each interval separately. Here, $\delta$ is a positive constant, and we set $\delta \le \ln\frac{\sqrt{\frac{1}{4L^2}+4}+\frac{1}{2L}}{2}$ to satisfy the condition of Lemma 23, which ensures the Lipschitz property of the score function, thereby allowing the discretization error to be bounded near the end of the data distribution.

Similar as Benton et al. (2024) and Chen et al. (2023a), we consider the step size satisfying $\eta_r \leq \eta \min(1, T - t_{r+1})$. $\eta$ is the parameters for controlling the maximum step size. We also assume there is an index $M$ and $N$ with $t_M = T - 1$ and $t_N = T - \delta$ as the work Benton et al. (2024). This assumption is purely for presentation clarity and our argument works similarly without it. To guarantee $\eta_r \leq \eta \min(1, T - t_{r+1})$, following the setting Benton et al. (2024), we first assume $M = \alpha R$ and $N = \beta R$ for convenience and further have

- At the time interval $[0, T - 1]$ where $t_0, \cdots, t_M$ are linearly spaced on $[0, T - 1]$, $\eta \geq \frac{T-1}{M} = \frac{T-1}{\alpha R}$ which can be satisfied by taking $\eta = \Omega\left(\frac{T}{R}\right)$.

- At the time interval $[T - 1, T - \delta]$ where $t_{M+1}, \cdots, t_N$ are exponentially decaying from $\frac{\eta}{1+\eta}$ to $\frac{\eta}{(1+\eta)^{N-M}}$. This condition can be satisfied with $\eta \geq (\frac{1}{\delta})^{\frac{1}{N-M}} - 1$. Assume $(\beta - \alpha)R \geq \log\frac{1}{\delta}$ and $e^{\frac{\ln\frac{1}{\delta}}{(\beta-\alpha)R}} \leq 1 + (e - 1)\frac{\ln\frac{1}{\delta}}{(\beta-\alpha)R}$, we can take $\eta = \Omega\left(\frac{\ln\frac{1}{\delta}}{R}\right)$.

- At the time interval $(T - \delta, T]$, the Lipschitz property of the score function at this time interval can be satisfied if we take $\delta \in \left(0, \ln\frac{\sqrt{\frac{1}{4L^2}+4}+\frac{1}{2L}}{2}\right]$.

Therefore, if we choose the constant $c = \ln\frac{\sqrt{\frac{1}{4L^2}+4}+\frac{1}{2L}}{2}$ and $\eta = \Theta\left(\frac{T+\ln\frac{1}{c}}{R}\right)$, we can have $\eta_r \leq \eta \min(1, T - t_{r+1})$ for $r = 0, \cdots, R - 1$ at time interval $[0, T]$. Then, we split the discretization error into three parts as

$$\sum_{r=0}^{R-1} \mathbb{E}_{\mathbf{y}_{[0:T]}^{\leftarrow}\sim Q_{[0:T]}^{\leftarrow}}\left[\int_{t_r}^{t_{r+1}} \left\|\nabla\ln q_{t_r}^{\leftarrow}(\mathbf{y}_{t_r}^{\leftarrow}) - \nabla\ln q_t^{\leftarrow}(\mathbf{y}_t^{\leftarrow})\right\|^2 \mathrm{d}t\right]$$

$$= \underbrace{\sum_{r=0}^{M-1} \mathbb{E}_{\mathbf{y}_{[0:T]}^{\leftarrow}\sim Q_{[0:T]}^{\leftarrow}}\left[\int_{t_r}^{t_{r+1}} \left\|\nabla\ln q_{t_r}^{\leftarrow}(\mathbf{y}_{t_r}^{\leftarrow}) - \nabla\ln q_t^{\leftarrow}(\mathbf{y}_t^{\leftarrow})\right\|^2 \mathrm{d}t\right]}_{\text{term I}}$$

$$+ \underbrace{\sum_{r=M}^{N-1} \mathbb{E}_{\mathbf{y}_{[0:T]}^{\leftarrow}\sim Q_{[0:T]}^{\leftarrow}}\left[\int_{t_r}^{t_{r+1}} \left\|\nabla\ln q_{t_r}^{\leftarrow}(\mathbf{y}_{t_r}^{\leftarrow}) - \nabla\ln q_t^{\leftarrow}(\mathbf{y}_t^{\leftarrow})\right\|^2 \mathrm{d}t\right]}_{\text{term II}}$$

$$+ \underbrace{\sum_{r=N}^{R-1} \mathbb{E}_{\mathbf{y}_{[0:T]}^{\leftarrow}\sim Q_{[0:T]}^{\leftarrow}}\left[\int_{t_r}^{t_{r+1}} \left\|\nabla\ln q_{t_r}^{\leftarrow}(\mathbf{y}_{t_r}^{\leftarrow}) - \nabla\ln q_t^{\leftarrow}(\mathbf{y}_t^{\leftarrow})\right\|^2 \mathrm{d}t\right]}_{\text{term III}}. \tag{39}$$

**Term I and Term II.** Following the proof of Lemma 2 in Benton et al. (2024). We first define our target as

$$E_{s,t} := \mathbb{E}_{\mathbf{y}_{[0:T]}^{\leftarrow}\sim Q_{[0:T]}^{\leftarrow}}\left[\|\nabla\ln q_t^{\leftarrow}(\mathbf{y}_t^{\leftarrow}) - \nabla\ln q_s^{\leftarrow}(\mathbf{y}_s^{\leftarrow})\|^2\right]$$

where $0 \leq s \leq t \leq T$.

By Lemma 14, we obtain

$$\frac{\mathrm{d}E_{s,t}}{\mathrm{d}t} = \frac{\mathrm{d}}{\mathrm{d}t}\mathbb{E}_{Q^{\leftarrow}}[\|\nabla\ln q_t^{\leftarrow}(\mathbf{y}_t^{\leftarrow})\|^2] - 2\frac{\mathrm{d}}{\mathrm{d}t}\mathbb{E}_{Q^{\leftarrow}}[\nabla\ln q_t^{\leftarrow}(\mathbf{y}_t^{\leftarrow})\cdot\nabla\ln \mathbf{y}_s^{\leftarrow}(\mathbf{y}_s^{\leftarrow})]$$

$$= 2\mathbb{E}_{Q^{\leftarrow}}[\|\nabla^2\ln q_t^{\leftarrow}(\mathbf{y}_t^{\leftarrow})\|_F^2] - 2\mathbb{E}_{Q^{\leftarrow}}[\|\nabla\ln q_t^{\leftarrow}(\mathbf{y}_t^{\leftarrow})\|^2] + 2\mathbb{E}_{Q^{\leftarrow}}[\nabla\ln q_t^{\leftarrow}(\tilde{\mathbf{y}}_t)\cdot\nabla\ln q_s^{\leftarrow}(\mathbf{y}_s^{\leftarrow})]$$

Using Young's inequality, we further have

$$\frac{\mathrm{d}E_{s,t}}{\mathrm{d}t} = 2\mathbb{E}_{Q^{\leftarrow}}[\|\nabla^2\ln q_t^{\leftarrow}(\tilde{\mathbf{y}}_t)\|_F^2] - 2\mathbb{E}_{Q^{\leftarrow}}[\|\nabla\ln q_t^{\leftarrow}(\mathbf{y}_t^{\leftarrow})\|^2] + 2\mathbb{E}_{Q^{\leftarrow}}[\nabla\ln q_t^{\leftarrow}(\mathbf{y}_t^{\leftarrow})\cdot\nabla\ln q_s^{\leftarrow}(\mathbf{y}_s^{\leftarrow})]$$

$$\leq \underbrace{\mathbb{E}_{Q^{\leftarrow}}[\|\nabla\ln q_s^{\leftarrow}(\mathbf{y}_s^{\leftarrow})\|^2]}_{\text{Term 1}} + \underbrace{2\mathbb{E}_{Q^{\leftarrow}}[\|\nabla^2\ln q_t^{\leftarrow}(\mathbf{y}_t^{\leftarrow})\|_F^2]}_{\text{Term 2}} \tag{40}$$

For Term 1, by Lemma 15, we can further bound Eq. 40 as

$$\mathbb{E}_{\mathbf{y}_{[0:T]} \sim Q_{[0:T]}}[\|\nabla \ln q_{T-s}(\mathbf{y}_{T-s})\|^2]$$

$$= \sigma_{T-s}^{-4} \mathbb{E}_Q[\|\mathbf{y}_{T-s}\|^2] - 2\sigma_{T-s}^{-4} e^{s-T} \mathbb{E}_Q[\mathbf{y}_{T-s} \cdot \boldsymbol{\mu}_{T-s}] + e^{-2(T-s)} \sigma_{T-s}^{-4} \mathbb{E}_Q[\|\boldsymbol{\mu}_{T-s}\|^2]$$

$$= \sigma_{T-s}^{-4} \mathbb{E}_Q[\|e^{s-T}\mathbf{y}_0 + \sigma_{T-s}\mathbf{z}\|^2] - 2\sigma_{T-s}^{-4} e^{s-T} \mathbb{E}_Q[\mathbf{y}_0 \cdot \mathbb{E}[\mathbf{y}_{T-s}|\mathbf{y}_0]] + e^{2(s-T)} \sigma_{T-s}^{-4} \mathbb{E}_Q[\|\boldsymbol{\mu}_{T-s}\|^2]$$

$$\le e^{2(s-T)} \sigma_{T-s}^{-4} \mathbb{E}_{\mathbf{y} \sim q_0} \left[\|\mathbf{y}\|^2\right] + \sigma_{T-s}^{-2} d - 2\sigma_{T-s}^{-4} e^{2(s-T)} \mathbb{E}_{\mathbf{y} \sim q_0} \left[\|\mathbf{y}\|^2\right] + e^{2(s-T)} \sigma_{T-s}^{-4} \mathbb{E}_Q[\|\boldsymbol{\mu}_{T-s}\|^2]$$

$$\overset{(a)}{=} \sigma_{T-s}^{-2} d + e^{2(s-T)} \sigma_{T-s}^{-4} \mathbb{E}_{\mathbf{y} \sim q_0} \left[\|\mathbf{y}\|^2\right] - 2\sigma_{T-s}^{-4} e^{2(s-T)} \mathbb{E}_{\mathbf{y} \sim q_0} \left[\|\mathbf{y}\|^2\right] + e^{2(s-T)} \sigma_{T-s}^{-4} [\mathbb{E}_{\mathbf{y} \sim q_0} \left[\|\mathbf{y}\|^2\right]$$

$$- \mathbb{E}_Q[\text{Tr}(\boldsymbol{\Sigma}_{T-s})]]$$

$$\le \sigma_{T-s}^{-2} d$$

where $\mathbf{z}$ is the standard Gaussian noise and Step (a) holds by the fact that $\text{Tr}(\boldsymbol{\Sigma}_t) = \mathbb{E}[\|\mathbf{y}_0\|^2|\mathbf{y}_t] - \mathbb{E}^2[\mathbf{y}_0|\mathbf{y}_t] = \mathbb{E}[\|\mathbf{y}_0\|^2|\mathbf{y}_t] - \|\boldsymbol{\mu}_t\|^2$ and $\mathbb{E}_{\tilde{Q}}[\text{Tr}(\boldsymbol{\Sigma}_t)] = \mathbb{E}_{p_{*,k+1|[1:k]}(\cdot|\boldsymbol{x}_{[1:k]})} \left[\|\mathbf{y}\|^2\right] - \mathbb{E}_{Q^{\leftarrow}}[\|\boldsymbol{\mu}_t\|^2]$. That is

$$\mathbb{E}_{Q^{\leftarrow}}[\|\nabla \ln q_s^{\leftarrow}(\mathbf{y}_s^{\leftarrow})\|^2] = \mathbb{E}_Q[\|\nabla \ln q_{T-s}(\mathbf{y}_{T-s})\|^2] \le \sigma_{T-s}^{-2} d$$

For Term 2, by Lemma 15, we expand Eq. 40 as

$$\mathbb{E}_{Q^{\leftarrow}}[\|\nabla^2 \ln q_t^{\leftarrow}(\tilde{\mathbf{y}}_t)\|_F^2] \tag{41}$$

$$= \sigma_{T-t}^{-4} d - 2\sigma_{T-t}^{-6} e^{-2(T-t)} \mathbb{E}_{Q^{\leftarrow}}[\text{Tr}(\boldsymbol{\Sigma}_{T-t})] + e^{-4(T-t)} \sigma_{T-t}^{-8} \mathbb{E}_{Q^{\leftarrow}}[\text{Tr}(\boldsymbol{\Sigma}_{T-t}^{\top} \boldsymbol{\Sigma}_{T-t})]$$

$$= \sigma_{T-t}^{-4} d - 2\sigma_{T-t}^{-6} e^{-2(T-t)} \mathbb{E}_{Q^{\leftarrow}}[\text{Tr}(\boldsymbol{\Sigma}_{T-t})] - \frac{1}{2} e^{-2(T-t)} \sigma_{T-t}^{-4} \frac{d}{dt} \mathbb{E}_{Q^{\leftarrow}}[\text{Tr}(\boldsymbol{\Sigma}_{T-t})]$$

$$\le \sigma_{T-t}^{-4} d - \frac{1}{2} \frac{d}{dt} \sigma_{T-t}^{-4} \mathbb{E}_{Q^{\leftarrow}}[\text{Tr}(\boldsymbol{\Sigma}_{T-t})] \tag{42}$$

The second equality holds by Lemma 25. The third inequality holds by the fact that $e^{-2(T-t)} \le 1$ and $\frac{d}{dt} \mathbb{E}_{\tilde{Q}}[\text{Tr}(\boldsymbol{\Sigma}_{T-t})] \le 0$ inferred by Lemma 25.

So, we can have the upper bound combined with two parts $E_{s,t}^{(1)}$ and $E_{s,t}^{(2)}$ as follows

$$\frac{dE_{s,t}}{dt} \le \underbrace{\sigma_{T-s}^{-2} d + 2\sigma_{T-t}^{-4} d}_{E_{s,t}^{(1)}} \underbrace{- \frac{d}{dr}(\sigma_{T-r}^{-4} \mathbb{E}_{Q^{\leftarrow}}[\text{Tr}(\boldsymbol{\Sigma}_{T-r})])|_{r=t}}_{E_{s,t}^{(2)}}$$

Therefore, when $s = t_r$, by Lemma 16, we have the upper bound of summation of term I and term II in Eq. 39 as

$$\sum_{r=0}^{N-1} \int_{t_r}^{t_{r+1}} E_{t_r,t} \le \sum_{r=0}^{N-1} \int_{t_r}^{t_{r+1}} \int_{t_r}^{t} E_{t_r,r}^{(1)} dr + \sum_{r=0}^{N-1} \int_{t_r}^{t_{r+1}} \int_{t_r}^{t} E_{t_r,r}^{(2)} dr$$

$$\lesssim dN\eta^2 + dT\eta + \mathbb{E}_{p_{*,k+1|[1:k]}(\cdot|\boldsymbol{x}_{[1:k]})} \left[\|\mathbf{y}\|^2\right] \eta. \tag{43}$$

Finally, we have

$$\text{Term I} + \text{Term II} = \sum_{r=0}^{N-1} \mathbb{E}_{\mathbf{y}_{[0:T]}^{\leftarrow} \sim Q_{[0:T]}^{\leftarrow}} \left[\int_{t_r}^{t_{r+1}} \left\|\nabla \ln q_{t_r}^{\leftarrow}(\mathbf{y}_{t_r}^{\leftarrow}) - \nabla \ln q_t^{\leftarrow}(\mathbf{y}_t^{\leftarrow})\right\|^2 dt\right]$$

$$\lesssim dN\eta^2 + dT\eta + \mathbb{E}_{p_{*,k+1|[1:k]}(\cdot|\boldsymbol{x}_{[1:k]})} \left[\|\mathbf{y}\|^2\right] \eta. \tag{44}$$

**Term III.** Since at this time interval, the corresponding forward process satisfies $\sigma_{0,t}^2 \leq \frac{\alpha_{0,t}}{2L}$, for $t \in (t_r, t_{r+1}]$, by Lemma 24, we have

$$\mathbb{E}_Q\left[\left\|\nabla \ln q_{T-t}(\mathbf{y}_{T-t}) - \nabla \ln q_{T-t_r}(\mathbf{y}_{T-t})\right\|^2\right]$$

$$\overset{(a)}{\leq} 4\mathbb{E}_Q\left\|\nabla \log q_{T-t}(\mathbf{y}_{T-t}) - \nabla \log q_{T-t}(\alpha_{T-t,T-t_r}^{-1}\mathbf{y}_{T-t_r})\right\|^2$$

$$+ 2\mathbb{E}_Q\left\|\nabla \log q_{T-t}(\mathbf{y}_{T-t})\right\|^2 \left(1 - \alpha_{T-t,T-t_r}^{-1}\right)^2$$

$$\overset{(b)}{\leq} 16L^2\alpha_{T-t}^{-2}\mathbb{E}_Q\|\mathbf{y}_{T-t} - \alpha_{T-t,T-t_r}^{-1}\mathbf{y}_{T-t_r}\|^2 + 2\mathbb{E}_Q\left\|\nabla \log q_{T-t}(\mathbf{y}_{T-t})\right\|^2 \left(1 - \alpha_{T-t,T-t_r}^{-1}\right)^2$$

$$\leq 16dL^2(e^{2(t-t_r)} - 1) + 2\mathbb{E}_Q\left\|\nabla \log q_{T-t}(\mathbf{y}_{T-t})\right\|^2 \left(1 - \alpha_{T-t,T-t_r}^{-1}\right)^2$$

$$\overset{(c)}{\leq} 32e^2dL^2(t - t_r) + 2\mathbb{E}_Q\left\|\nabla \log q_{T-t}(\mathbf{y}_{T-t})\right\|^2 (t - t_r)^2$$

The first inequality (a) holds by Lemma 24 and the second inequality (b) holds by Lemma 23. Step (c) holds by the assumption that the step size at the given time interval $t - t_r \leq \eta_r \leq 1$. Since $\nabla \log q_t$ is $2L\alpha_t^{-1}$-Lipschitz, we obtain

$$\mathbb{E}_Q\left\|\nabla \log q_{T-t}(\mathbf{y}_{T-t})\right\|^2 \tag{45}$$

$$= \int q_{T-t}(\mathbf{y}_{T-t})\left\|\nabla \log q_{T-t}(\mathbf{y}_{T-t})\right\|^2 d\mathbf{y}_{T-t}$$

$$= \int \langle \nabla q_{T-t}(\mathbf{y}_{T-t}), \nabla \log q_{T-t}(\mathbf{y}_{T-t})\rangle d\mathbf{y}_{T-t}$$

$$= \int q_{T-t}(\mathbf{y}_{T-t})\Delta \log q_{T-t}(\mathbf{y}_{T-t})d\mathbf{y}_{T-t}$$

$$\leq 2dL. \tag{46}$$

That is, we have

$$\mathbb{E}_{Q^{\leftarrow}}\left[\left\|\nabla \ln q_t^{\leftarrow}(\mathbf{y}_t^{\leftarrow}) - \nabla \ln q_{t_r}^{\leftarrow}(\mathbf{y}_{t_r}^{\leftarrow})\right\|^2\right] \leq 32e^2dL^2(t - t_r) + 4dL(t - t_r)^2. \tag{47}$$

Then, plug Eq. 45 into Eq. 47, for step sizesatisfies

$$\eta_r \leq \min\{1, \eta, \eta(T - t_{r+1})\},$$

we have

$$\text{Term III} = \sum_{r=N}^{R-1} \mathbb{E}_{\tilde{\mathbf{y}}_{[0:T]} \sim \tilde{Q}_{[0:T]}}\left[\int_{t_r}^{t_{r+1}} \|\nabla \ln \tilde{q}_{t_r}(\tilde{\mathbf{y}}_{t_r}) - \nabla \ln \tilde{q}_t(\tilde{\mathbf{y}}_t)\|^2 dt\right]$$

$$\lesssim dL^2 \sum_{r=N}^{R-1} h_k^2 \lesssim dL^2 R\eta^2. \tag{48}$$

Finally, combine Eq. 44 with 48, assume $L \geq 1$, the discretization error can be bounded as

$$\sum_{r=0}^{R-1} \mathbb{E}_{\mathbf{y}_{[0:T]}^{\leftarrow} \sim Q_{[0:T]}^{\leftarrow}}\left[\int_{t_r}^{t_{r+1}} \left\|\nabla \ln q_{t_r}^{\leftarrow}(\mathbf{y}_{t_r}^{\leftarrow}) - \nabla \ln q_t^{\leftarrow}(\mathbf{y}_t^{\leftarrow})\right\|^2 dt\right]$$

$$\lesssim dL^2 R\eta^2 + dT\eta + \mathbb{E}_{p_{*,k+1|[1:k]}(\cdot|\boldsymbol{x}_{[1:k]})}\left[\|\mathbf{y}\|^2\right]\eta.$$

$\square$

**Lemma 13.** *If* $\mathbf{y}_t^{\leftarrow}$ *is the solution to SDE Equation (4), then for all* $t \in [0, T]$, *we have*

$$d(\nabla \ln q_t^{\leftarrow}(\mathbf{y}_t^{\leftarrow})) = \sqrt{2}\nabla^2 \ln q_t^{\leftarrow}(\mathbf{y}_t^{\leftarrow})d\boldsymbol{B}_t - \nabla \log q_t^{\leftarrow}(\mathbf{y}_t^{\leftarrow})dt \tag{49}$$

*Proof.* Since $\nabla \ln q_{t_r}^{\leftarrow}(\mathbf{y}_{t_r}^{\leftarrow})$ in process 4 is smooth for $t \in [0, T]$, by Itô lemma, we expand

$$d(\nabla \ln q_t^{\leftarrow}(\mathbf{y}_t^{\leftarrow}))$$

$$= \nabla^2 \ln q_t^{\leftarrow}(\mathbf{y}_t^{\leftarrow})d\mathbf{y}_t^{\leftarrow} + \frac{d(\nabla \ln q_t^{\leftarrow}(\mathbf{y}_t^{\leftarrow}))}{dt}dt + \frac{1}{2} \cdot (\sqrt{2})^2 \Delta(\nabla \ln q_t^{\leftarrow}(\mathbf{y}_t^{\leftarrow}))dt$$

$$= \Big\{ \nabla^2 \ln q_t^{\leftarrow}(\mathbf{y}_t^{\leftarrow}) \cdot ((\mathbf{y}_t^{\leftarrow} + 2\nabla \ln q_{T-t}(\mathbf{y}_t^{\leftarrow})) + \Delta(\nabla \ln q_t^{\leftarrow}(\mathbf{y}_t^{\leftarrow})) \Big\}dt + \sqrt{2}\nabla^2 \ln q_t^{\leftarrow}(\mathbf{y}_t^{\leftarrow})d\mathbf{B}_t \tag{50}$$

$$+ d(\nabla \ln q_t^{\leftarrow}(\mathbf{y}_t^{\leftarrow})). \tag{51}$$

By Fokker–Planck Eq. for the forward process 3, we have

$$\frac{dq_t(\mathbf{y}_t)}{dt} = \nabla \cdot (\mathbf{y}_t q_t(\mathbf{y}_t)) + \Delta q_t(\mathbf{y}_t).$$

We define $f(\mathbf{y}_t) = \log q_t(\mathbf{y}_t)$ and $q_t(\mathbf{y}_t) = \exp(f)$, thus

$$\frac{dq_t(\mathbf{y}_t)}{dt} = q_t(\mathbf{y}_t)\frac{df(\mathbf{y}_t)}{dt}$$

$$= \nabla \cdot (\mathbf{y}_t q_t(\mathbf{y}_t)) + \Delta q_t(\mathbf{y}_t)$$

$$= q_t(\mathbf{y}_t)(d + \mathbf{y}_t \nabla f(\mathbf{y}_t)) + q_t(\mathbf{y}_t)(\|\nabla f(\mathbf{y}_t)\|^2 + \Delta f(\mathbf{y}_t))$$

$$= q_t(\mathbf{y}_t)(d + \mathbf{y}_t \nabla \log q_t(\mathbf{y}_t) + q_t(\mathbf{y}_t)(\|\nabla \log q_t(\mathbf{y}_t)\|^2 + \Delta \log q_t(\mathbf{y}_t))$$

So, back to the reverse process 4 where $t \mapsto T - t$

$$d\nabla \log q_{T-t}(\mathbf{y}_{T-t})$$

$$= -\nabla\Big(d + \mathbf{y}_{T-t} \cdot \nabla \log q_{T-t}(\mathbf{y}_{T-t}) + \|\nabla \log q_{T-t}(\mathbf{y}_{T-t})\|^2 + \Delta \log q_{T-t}(\mathbf{y}_{T-t})\Big)$$

$$= -\Big(\nabla \log q_{T-t}(\mathbf{y}_{T-t}) + \mathbf{y}_{T-t} \cdot \nabla^2 \log q_{T-t}(\mathbf{y}_{T-t}) + 2\nabla \log q_{T-t}(\mathbf{y}_{T-t}) \cdot \nabla^2 \log q_{T-t}(\mathbf{y}_{T-t})$$

$$+ \nabla(\Delta \log q_{T-t}(\mathbf{y}_{T-t}))\Big)dt.$$

It means, for the reverse process, we have

$$d\nabla \log q_t^{\leftarrow}(\mathbf{y}_t^{\leftarrow}) \tag{52}$$

$$= -\Big(\nabla \log q_t^{\leftarrow}(\mathbf{y}_t^{\leftarrow}) + \mathbf{y}_t^{\leftarrow} \cdot \nabla^2 \log q_t^{\leftarrow}(\mathbf{y}_t^{\leftarrow}) + 2\nabla q_t^{\leftarrow}(\mathbf{y}_t^{\leftarrow}) \cdot \nabla^2 q_t^{\leftarrow}(\mathbf{y}_t^{\leftarrow}) + \nabla(\Delta q_t^{\leftarrow}(\mathbf{y}_t^{\leftarrow}))\Big)dt. \tag{53}$$

Plugging Eq. 52 into 50, we have

$$d(\nabla \ln q_t^{\leftarrow}(\mathbf{y}_t^{\leftarrow})) = \sqrt{2}\nabla^2 \ln q_t^{\leftarrow}(\mathbf{y}_t^{\leftarrow})d\mathbf{B}_t - \nabla \log q_t^{\leftarrow}(\mathbf{y}_t^{\leftarrow})dt$$

The proof is complete. $\qquad \square$

**Lemma 14.** *If $\mathbf{y}_t^{\leftarrow}$ is the solution to SDE Equation* (4)*, then for all $t \in [0, T]$, we have*

$$2\mathbb{E}_{Q^{\leftarrow}}[\|\nabla \ln q_t^{\leftarrow}(\mathbf{y}_t^{\leftarrow})\|^2] + \frac{d\mathbb{E}_{Q^{\leftarrow}}[\|\nabla \ln q_t^{\leftarrow}(\mathbf{y}_t^{\leftarrow})\|^2]}{dt} = 2\mathbb{E}_{Q^{\leftarrow}}[\|\nabla^2 \ln q_t^{\leftarrow}(\mathbf{y}_t^{\leftarrow})\|_F^2].$$

*Proof.* By Lemma 13, we have

$$d(e^t \nabla \ln q_t^{\leftarrow}(\mathbf{y}_t^{\leftarrow})) = \sqrt{2}e^t\nabla^2 \ln q_t^{\leftarrow}(\mathbf{y}_t^{\leftarrow})d\mathbf{B}_t.$$

By Eq. 41, we know $\int_s^2 \mathbb{E}_{Q^{\leftarrow}}[\nabla^2 \ln q_t^{\leftarrow}(\mathbf{y}_t^{\leftarrow})] \leq \infty$ which is square integrability and then applying Itô isometry, for $0 \leq s \leq t \leq T$, we obtain

$$\frac{d}{dt}\mathbb{E}_{Q^{\leftarrow}}[\|e^t \nabla \ln q_t^{\leftarrow}(\mathbf{y}_t^{\leftarrow}) - e^s \nabla \ln q_s^{\leftarrow}(\mathbf{y}_s^{\leftarrow})\|^2] = 2e^{2t}\mathbb{E}_{Q^{\leftarrow}}[\|\nabla^2 \ln q_t^{\leftarrow}(\mathbf{y}_t^{\leftarrow})\|_F^2]. \tag{54}$$

where $\|\mathbf{A}\|_F^2 = \text{Tr}(\mathbf{A}^\top \mathbf{A})$ is squared Frobenius norm of a matrix. We further expand Eq. 54 as

$$2\mathbb{E}_{Q^{\leftarrow}}[\|\nabla \ln q_t^{\leftarrow}(\mathbf{y}_t^{\leftarrow})\|^2] + \frac{d\mathbb{E}_{\tilde{Q}}[\|\nabla \ln q_t^{\leftarrow}(\mathbf{y}_t^{\leftarrow})\|^2]}{dt} - 2e^{s-t}\mathbb{E}_{Q^{\leftarrow}}[\nabla \ln q_t^{\leftarrow}(\mathbf{y}_t^{\leftarrow}) \cdot \nabla \ln q_s^{\leftarrow}(\mathbf{y}_s^{\leftarrow})]$$

$$- 2e^{s-t}\frac{d}{dt}\mathbb{E}_{Q^{\leftarrow}}[\nabla \ln q_t^{\leftarrow}(\mathbf{y}_t^{\leftarrow}) \cdot \nabla \ln q_s^{\leftarrow}(\mathbf{y}_s^{\leftarrow})] = 2\mathbb{E}_{Q^{\leftarrow}}[\|\nabla^2 \ln q_t^{\leftarrow}(\mathbf{y}_t^{\leftarrow})\|_F^2]. \tag{55}$$

Given any $s$, we have $t \geq s$,

$$d(\nabla \ln q_s^{\leftarrow}(\mathbf{y}_s^{\leftarrow}) \cdot \nabla \ln q_t^{\leftarrow}(\mathbf{y}_t^{\leftarrow})) = \sqrt{2}\nabla \ln q_s^{\leftarrow}(\mathbf{y}_s^{\leftarrow}) \cdot \nabla^2 \ln q_t^{\leftarrow}(\mathbf{y}_t^{\leftarrow})d\mathbf{B}_t$$
$$- \nabla \ln q_s^{\leftarrow}(\mathbf{y}_s^{\leftarrow}) \cdot \nabla \log q_t^{\leftarrow}(\mathbf{y}_t^{\leftarrow})dt.$$

And by Eq. 41, we know $\int_s^2 \mathbb{E}_{Q^{\leftarrow}}[\nabla^2 \ln q_t^{\leftarrow}(\mathbf{y}_t^{\leftarrow})] \leq \infty$ which is square integrability, applying Itô isometry take the expectation at both side, we take

$$\frac{d}{dt}\mathbb{E}_{Q^{\leftarrow}}[\nabla \ln q_s^{\leftarrow}(\mathbf{y}_s^{\leftarrow}) \cdot \nabla \ln q_t^{\leftarrow}(\mathbf{y}_t^{\leftarrow})] = -\mathbb{E}_{Q^{\leftarrow}}[\nabla \ln q_s^{\leftarrow}(\mathbf{y}_s^{\leftarrow}) \cdot \nabla \log q_t^{\leftarrow}(\mathbf{y}_t^{\leftarrow})].$$

Therefore, Eq. 55 can be simplified as

$$2\mathbb{E}_{Q^{\leftarrow}}[\|\nabla \ln q_t^{\leftarrow}(\mathbf{y}_t^{\leftarrow})\|^2] + \frac{d\mathbb{E}_{\tilde{Q}}[\|\nabla \ln q_t^{\leftarrow}(\mathbf{y}_t^{\leftarrow})\|^2]}{dt} = 2\mathbb{E}_{Q^{\leftarrow}}[\|\nabla^2 \ln q_t^{\leftarrow}(\mathbf{y}_t^{\leftarrow})\|_F^2].$$

$\square$

**Lemma 15.** *For all $t > 0$, we have*

$$\nabla \log q_t^{\leftarrow}(\mathbf{y}_t^{\leftarrow}) = \nabla \log q_{T-t}(\mathbf{y}_{T-t}) = \sigma_{T-t}^{-2}\mathbf{y}_t^{\leftarrow} - e^{t-T}\sigma_{T-t}^{-2}\boldsymbol{\mu}_{T-t},$$
$$\nabla^2 \log q_t^{\leftarrow}(\mathbf{y}_t^{\leftarrow}) = \nabla^2 \log q_{T-t}(\mathbf{y}_{T-t}) = -\sigma_{T-t}^{-2}I + e^{-2(T-t)}\sigma_{T-t}^{-4}\boldsymbol{\Sigma}_{T-t}.$$

*Proof.* For the forward process 3, we have

$$\nabla \ln q_t(\mathbf{y}_t) = \frac{1}{q_t(\mathbf{y}_t)}\int \nabla \log q_{t|0}(\mathbf{y}_t|\mathbf{y}_0)q_{0,t}(\mathbf{y}_0, \mathbf{y}_t)d\mathbf{y}_0$$
$$= -\mathbb{E}_{q_{0|t}}[\sigma_t^{-2}(\mathbf{y}_t - \alpha_{0,t}\mathbf{y}_0)]$$
$$= -\sigma_t^{-2}\mathbf{y}_t + e^{-t}\sigma_t^{-2}\boldsymbol{\mu}_t$$

where the last equality holds by the forward process $q_{t|0}(\mathbf{y}_t|\mathbf{y}_0) = \mathcal{N}(\mathbf{y}_t; \alpha_{0,t}\mathbf{y}_0, \sigma_t^2 I)$ and $\nabla \log q_{t|0}(\mathbf{y}_t|\mathbf{y}_0) = -\sigma_t^{-2}(\mathbf{y}_t - \alpha_{0,t}\mathbf{y}_0)$.

Then, the second-order derivative is

$$\nabla^2 \log q_t(\mathbf{y}_t)$$
$$= \frac{1}{q_t(\mathbf{y}_t)}\int \nabla^2 \log q_{t|0}(\mathbf{y}_t|\mathbf{y}_0)q_{0,t}(\mathbf{y}_0, \mathbf{y}_t)d\mathbf{y}_0$$
$$+ \frac{1}{q_t(\mathbf{y}_t)}\int (\nabla \log q_{t|0}(\mathbf{y}_t|\mathbf{y}_0))(\nabla \log q_{t|0}(\mathbf{y}_t|\mathbf{y}_0))^{\top}q_{0,t}(\mathbf{y}_0, \mathbf{y}_t)d\mathbf{y}_0$$
$$- \frac{1}{q_t(\mathbf{y}_t)^2}\left(\int \nabla \log q_{t|0}(\mathbf{y}_t|\mathbf{y}_0)q_{0,t}(\mathbf{y}_0, \mathbf{y}_t)d\mathbf{y}_0\right)\left(\int \nabla \log q_{t|0}(\mathbf{y}_t|\mathbf{y}_0)q_{0,t}(\mathbf{y}_0, \mathbf{y}_t)d\mathbf{y}_0\right)^{\top}$$
$$= -\frac{1}{\sigma_t^2}I + \mathbb{E}_{q_{0|t}(\cdot|\mathbf{y}_t)}[\sigma_t^{-4}(\mathbf{y}_t - \mathbf{y}_0 e^{-t})(\mathbf{y}_t - \mathbf{y}_0 e^{-t})^{\top}]$$
$$- \mathbb{E}_{q_{0|t}(\cdot|\mathbf{y}_t)}[-\sigma_t^{-2}(\mathbf{y}_t - \mathbf{y}_0 e^{-t})]\mathbb{E}_{q_{0|t}(\cdot|\mathbf{y}_t)}[-\sigma_t^{-2}(\mathbf{y}_t - \mathbf{y}_0 e^{-t})]^{\top}$$
$$= -\sigma_t^{-2}I + \sigma_t^{-4}\mathrm{Cov}_{q_{0|t}(\cdot|\mathbf{y}_t)}(\mathbf{y}_t - \mathbf{y}_0 e^{-t})$$
$$= -\sigma_t^{-2}I + e^{-2t}\sigma_t^{-4}\boldsymbol{\Sigma}_t.$$

Therefore,

$$\nabla \log q_t^{\leftarrow}(\mathbf{y}_t^{\leftarrow}) = \nabla \log q_{T-t}(\mathbf{y}_{T-t}) = \sigma_{T-t}^{-2}\tilde{\mathbf{y}}_t - e^{t-T}\sigma_{T-t}^{-2}\boldsymbol{\mu}_{T-t},$$
$$\nabla^2 \log q_t^{\leftarrow}(\mathbf{y}_t^{\leftarrow}) = \nabla^2 \log q_{T-t}(\mathbf{y}_{T-t}) = -\sigma_{T-t}^{-2}I + e^{-2(T-t)}\sigma_{T-t}^{-4}\boldsymbol{\Sigma}_{T-t}.$$

$\square$

**Lemma 16.** *Define*

$$\frac{dE_{s,t}}{dt} \leq \underbrace{\sigma_{T-s}^{-2}d + 2\sigma_{T-t}^{-4}d}_{E_{s,t}^{(1)}} \underbrace{-\frac{d}{dr}(\sigma_{T-r}^{-4}\mathbb{E}_{\tilde{Q}}[Tr(\boldsymbol{\Sigma}_{T-r})])|_{r=t}}_{E_{s,t}^{(2)}}.$$

The error terms $E_{s,t}^{(1)}$ and $E_{s,t}^{(2)}$ satisfies

$$\sum_{k=M}^{N-1} \int_{t_r}^{t_{r+1}} \int_{t_r}^{t} E_{t_r,r}^{(1)} \mathrm{d}r \mathrm{d}t \lesssim dN\eta^2,$$

$$\sum_{r=0}^{N-1} \int_{t_r}^{t_{r+1}} \int_{t_r}^{t} E_{t_r,r}^{(2)} \mathrm{d}r \mathrm{d}t \lesssim \eta \mathbb{E}_{p_{*,k+1|[1:k]}(\cdot|\boldsymbol{x}_{[1:k]})}\left[\|\mathbf{y}\|^2\right] + dN\eta^2$$

*Proof.* We first define

$$\frac{\mathrm{d}E_{s,t}}{\mathrm{d}t} \leq \underbrace{\sigma_{T-s}^{-2}d + 2\sigma_{T-t}^{-4}d}_{E_{s,t}^{(1)}} \underbrace{- \frac{\mathrm{d}}{\mathrm{d}r}(\sigma_{T-r}^{-4}\mathbb{E}_{\tilde{Q}}[\mathrm{Tr}(\boldsymbol{\Sigma}_{T-r})])|_{r=t}}_{E_{s,t}^{(2)}}$$

For $E_{s,t}^{(1)}$, consider the time interval $s, t \in [0, T-1]$ and $\sigma_{T-s}^2 \geq \sigma_{T-t}^2 = 1 - e^{-2(T-t)} \geq \frac{4}{5}$, we derive

$$\sum_{k=0}^{M-1} \int_{t_r}^{t_{r+1}} \int_{t_r}^{t} E_{t_r,r}^{(1)} \mathrm{d}r \mathrm{d}t \lesssim d \sum_{k=0}^{k=M} h_k^2 \lesssim \eta dT$$

Consider the time interval $(T-1, T-\delta]$, $\frac{1}{5}(T-t) \leq \sigma_{T-t}^2 \leq \sigma_{T-s}^2 \leq 2(T-s)$, we derive

$$\sum_{k=M}^{N-1} \int_{t_r}^{t_{r+1}} \int_{t_r}^{t} E_{t_r,r}^{(1)} \mathrm{d}r \mathrm{d}t$$

$$\lesssim d \sum_{k=M}^{N-1} \int_{t_r}^{t_{r+1}} \int_{t_r}^{t} (T-r)^{-2} \mathrm{d}r$$

$$\lesssim d \sum_{k=M}^{N-1} \frac{h_k^2}{(T-t_{r+1})^2}$$

$$\lesssim dN\eta^2.$$

The last inequality holds by the setting $\eta_r \leq \eta \min(1, T - t_{r+1})$.

For $E_{s,t}^{(2)}$, we first have

$$\sum_{r=0}^{N-1} \int_{t_r}^{t_{r+1}} \int_{t_r}^{t} E_{t_r,r}^{(2)} \mathrm{d}r \mathrm{d}t = \sum_{r=0}^{N-1} \int_{t_r}^{t_{r+1}} \left( \sigma_{T-t_r}^{-4} \mathbb{E}_{Q\leftarrow}[\mathrm{Tr}(\boldsymbol{\Sigma}_{T-t_r})] - \sigma_{T-t}^{-4}\mathbb{E}_{Q\leftarrow}[\mathrm{Tr}(\boldsymbol{\Sigma}_{T-t})] \right) \mathrm{d}t$$

$$\overset{(a)}{\leq} \sum_{r=0}^{N-1} \eta_r \sigma_{T-t_r}^{-4} \left( \mathbb{E}_{Q\leftarrow}[\mathrm{Tr}(\boldsymbol{\Sigma}_{T-t_r})] - \mathbb{E}_{Q\leftarrow}[\mathrm{Tr}(\boldsymbol{\Sigma}_{T-t_{r+1}})] \right)$$

Step (a) holds by the fact that $\sigma_{T-t}^{-4}$ is increasing with $t$ and $\mathbb{E}_{Q\leftarrow}[\mathrm{Tr}(\boldsymbol{\Sigma}_{T-t})]$ is decreasing with $t$ ($\frac{\mathrm{d}}{\mathrm{d}t}\mathbb{E}_{\tilde{Q}}[\mathrm{Tr}(\boldsymbol{\Sigma}_{T-t})] \leq 0$ inferred from Lemma 25).

Then, consider the time interval $[0, T-1]$ and $\eta_r \sigma_{T-t_r}^{-4} \leq \frac{25}{16}\eta_r \leq \frac{25}{16}\eta$, thus

$$\sum_{k=0}^{M-1} \eta_r \sigma_{T-t_r}^{-4} \left( \mathbb{E}_{Q\leftarrow}[\mathrm{Tr}(\boldsymbol{\Sigma}_{T-t_r})] - \mathbb{E}_{Q\leftarrow}[\mathrm{Tr}(\boldsymbol{\Sigma}_{T-t_{r+1}})] \right)$$

$$\leq \frac{25}{16}\eta \sum_{k=0}^{M-1} \left( \mathbb{E}_{Q\leftarrow}[\mathrm{Tr}(\boldsymbol{\Sigma}_{T-t_r})] - \mathbb{E}_{Q\leftarrow}[\mathrm{Tr}(\boldsymbol{\Sigma}_{T-t_{r+1}})] \right)$$

$$\leq \frac{25}{16}\eta \mathbb{E}_{Q\leftarrow}[\mathrm{Tr}(\boldsymbol{\Sigma}_T)]$$

$$\lesssim \eta \mathbb{E}_{\mathbf{y}\sim q_0}\left[\|\mathbf{y}\|^2\right].$$

The last inequality holds by $\mathbb{E}_{Q\leftarrow}[\text{Tr}(\boldsymbol{\Sigma}_T)] \leq \mathbb{E}_{Q\leftarrow}[\mathbb{E}[\|\mathbf{y}_0\|^2|\mathbf{y}_t]] \leq \mathbb{E}_{\mathbf{y}\sim q_0}\left[\|\mathbf{y}\|^2\right]$.

Similarly, consider the time interval $(T-1, T-\delta]$, $\eta_r \sigma_{T-t_r}^{-4} \leq \frac{25}{(T-t_r)^2}\eta_r \leq \frac{25}{(T-t_r)}\eta$, then

$$\sum_{k=M}^{N-1} \eta_r \sigma_{T-t_r}^{-4} \left( \mathbb{E}_{Q\leftarrow}[\text{Tr}(\boldsymbol{\Sigma}_{T-t_r})] - \mathbb{E}_{Q\leftarrow}[\text{Tr}(\boldsymbol{\Sigma}_{T-t_{r+1}})]\right) \tag{56}$$

$$\leq \frac{25}{16}\eta \sum_{k=M}^{N-1} \frac{1}{T-t_r}\left( \mathbb{E}_{Q\leftarrow}[\text{Tr}(\boldsymbol{\Sigma}_{T-t_r})] - \mathbb{E}_{Q\leftarrow}[\text{Tr}(\boldsymbol{\Sigma}_{T-t_{r+1}})]\right)$$

$$\leq \frac{25}{16}\eta\mathbb{E}_{Q\leftarrow}[\text{Tr}(\boldsymbol{\Sigma}_1)] + \frac{25}{16}\eta \sum_{k=M+1}^{N-1} \frac{\eta_{r-1}}{(T-t_r)(T-t_{k-1})} \mathbb{E}_{Q\leftarrow}[\text{Tr}(\boldsymbol{\Sigma}_{T-t_r})]$$

$$\leq \frac{25}{16}\eta\mathbb{E}_{Q\leftarrow}[\text{Tr}(\boldsymbol{\Sigma}_1)] + \frac{25}{16}\eta^2 \sum_{k=M+1}^{N-1} \frac{1}{T-t_{k-1}} \mathbb{E}_{Q\leftarrow}[\text{Tr}(\boldsymbol{\Sigma}_{T-t_r})] \tag{57}$$

For the interval $t \in (T-1, T-\delta]$, we have

$$\mathbb{E}_{Q\leftarrow}[\text{Tr}(\boldsymbol{\Sigma}_{T-t})] = \mathbb{E}_{Q\leftarrow}[\text{Tr}(\text{Cov}(e^{T-t}\mathbf{y}_{T-t} - \mathbf{y}_0|\mathbf{y}_{T-t}))]$$

$$= e^{2(T-t)}\mathbb{E}_{Q\leftarrow}[\text{Tr}(\text{Cov}(\mathbf{y}_{T-t} - e^{t-T}\mathbf{y}_0|\mathbf{y}_{T-t}))]$$

$$\leq e^{2(T-t)}\mathbb{E}_{Q\leftarrow}[\mathbb{E}[\|\mathbf{y}_{T-t} - e^{t-T}\mathbf{y}_0\|^2|\mathbf{y}_{T-t}]]$$

$$\leq d(e^{2(T-t)} - 1)$$

$$\leq 2de^2(T-t) \tag{58}$$

Then, plugging Eq. 58 into Eq. 56, we have

$$\sum_{k=M}^{N-1} \eta_r \sigma_{T-t_r}^{-4} \left( \mathbb{E}_{Q\leftarrow}[\text{Tr}(\boldsymbol{\Sigma}_{T-t_r})] - \mathbb{E}_{Q\leftarrow}[\text{Tr}(\boldsymbol{\Sigma}_{T-t_{r+1}})]\right)$$

$$\leq \frac{25}{16}\eta\mathbb{E}_{p_{*,k+1|[1:k]}(\cdot|\boldsymbol{x}_{[1:k]})}\left[\|\mathbf{y}\|^2\right] + \frac{25de^2}{8}\eta^2 \sum_{k=M+1}^{N-1} \frac{T-t_r}{T-t_{k-1}}$$

$$\lesssim \eta\mathbb{E}_{p_{*,k+1|[1:k]}(\cdot|\boldsymbol{x}_{[1:k]})}\left[\|\mathbf{y}\|^2\right] + dN\eta^2$$

Therefore, we obtain

$$\sum_{r=0}^{N-1} \int_{t_r}^{t_{r+1}} \int_{t_r}^{t} E_{t_r,r}^{(2)}\mathrm{d}r\mathrm{d}t \lesssim \eta\mathbb{E}_{p_{*,k+1|[1:k]}(\cdot|\boldsymbol{x}_{[1:k]})}\left[\|\mathbf{y}\|^2\right] + dN\eta^2$$

$\square$

## C  LOWER BOUND FOR VANILLA DIFFUSION MODELS

*Proof of Lemma 5.* Denote $d_{\boldsymbol{x}} = d_1 + d_2 + \cdots + d_k$, and $d_{\mathbf{y}} = d_{k+1}$, and denote $d = d_{\boldsymbol{x}} + d_{\mathbf{y}}$. Choose $p_*(\mathbf{y}, \boldsymbol{x})$ to be the density function of $\mathcal{N}\left(0, \begin{pmatrix} \mathrm{I}_{d_{\mathbf{y}}\times d_{\mathbf{y}}} & \frac{\epsilon}{dM}\bar{\mathrm{I}}_{d_{\mathbf{y}}\times d_{\boldsymbol{x}}} \\ \frac{\epsilon}{dM}\bar{\mathrm{I}}_{d_{\boldsymbol{x}}\times d_{\mathbf{y}}} & 2\left(\frac{\epsilon^2}{d^2M^2} + \frac{\epsilon^3}{d^2M^2}\right)\mathrm{I}_{d_{\boldsymbol{x}}\times d_{\boldsymbol{x}}} \end{pmatrix}\right)$, and $\hat{p}_*(\mathbf{y}, \boldsymbol{x})$ the density function of $\mathcal{N}\left(0, \begin{pmatrix} \mathrm{I}_{d_{\mathbf{y}}\times d_{\mathbf{y}}} & \frac{\epsilon}{dM}\bar{\mathrm{I}}_{d_{\mathbf{y}}\times d_{\boldsymbol{x}}} \\ \frac{\epsilon}{dM}\bar{\mathrm{I}}_{d_{\boldsymbol{x}}\times d_{\mathbf{y}}} & 2\frac{\epsilon^2}{d^2M^2}\mathrm{I}_{d_{\boldsymbol{x}}\times d_{\boldsymbol{x}}} \end{pmatrix}\right)$.

$\hat{p}_*(\mathbf{y}|\boldsymbol{x})$ corresponds to $\mathcal{N}\left(\frac{dM}{2\epsilon}\bar{\mathrm{I}}_{d_{\mathbf{y}}\times d_{\boldsymbol{x}}}\boldsymbol{x}, \frac{1}{2}\mathrm{I}_{d_{\mathbf{y}}\times d_{\mathbf{y}}}\right)$, and $p_*(\mathbf{y}|\boldsymbol{x})$ corresponds to $\mathcal{N}\left(\frac{dM}{2\epsilon}\frac{1}{1+\epsilon}\bar{\mathrm{I}}_{d_{\mathbf{y}}\times d_{\boldsymbol{x}}}\boldsymbol{x}, \frac{1}{2}\left(1 + \frac{\epsilon}{1+\epsilon}\right)\mathrm{I}_{d_{\mathbf{y}}\times d_{\mathbf{y}}}\right)$. Hence $\text{KL}\left(p_*(\mathbf{y}|\boldsymbol{x})\|\hat{p}_*(\mathbf{y}|\boldsymbol{x})\right) > \left(\frac{M}{1/d+\epsilon/d}\right)^2\left\|\bar{\mathrm{I}}_{d_{\mathbf{y}}\times d_{\boldsymbol{x}}}\boldsymbol{x}\right\|^2 = \left(\frac{M}{1/d+\epsilon/d}\right)^2\left\|\boldsymbol{x}_{(1:d_{k+1})}\right\|^2$.

On the other hand,

$$
\begin{pmatrix} \mathrm{I}_{d_{\mathbf{y}} \times d_{\mathbf{y}}} & \frac{\epsilon}{dM} \bar{\mathrm{I}}_{d_{\mathbf{y}} \times d_{\boldsymbol{x}}} \\ \frac{\epsilon}{dM} \bar{\mathrm{I}}_{d_{\boldsymbol{x}} \times d_{\mathbf{y}}} & 2 \frac{\epsilon^2}{d^2 M^2} \mathrm{I}_{d_{\boldsymbol{x}} \times d_{\boldsymbol{x}}} \end{pmatrix}^{-1}
$$
$$
= \frac{d^2 M^2}{\epsilon^2} \begin{pmatrix} 2 \frac{\epsilon^2}{d^2 M^2} \mathrm{I}_{d_{\mathbf{y}} \times d_{\mathbf{y}}} & -\frac{\epsilon}{dM} \bar{\mathrm{I}}_{d_{\mathbf{y}} \times d_{\boldsymbol{x}}} \\ -\frac{\epsilon}{dM} \bar{\mathrm{I}}_{d_{\boldsymbol{x}} \times d_{\mathbf{y}}} & \frac{1}{2} \mathrm{I}_{d_{\boldsymbol{x}} \times d_{\boldsymbol{x}}} + \frac{1}{2} \bar{\mathrm{I}}_{d_{\boldsymbol{x}} \times d_{\mathbf{y}}} \bar{\mathrm{I}}_{d_{\mathbf{y}} \times d_{\boldsymbol{x}}} \end{pmatrix}
$$

Hence $\mathrm{KL}\left(p_*(\mathbf{y}, \boldsymbol{x}) \big\| \hat{p}_*(\mathbf{y}, \boldsymbol{x})\right) \le \epsilon(d_x + d_y) = \epsilon d$.

Choosing $\varepsilon = \epsilon d$, and noting that $d \ge 2$ and that $\varepsilon \le 1/2$ finishes the proof. $\qquad \square$

## D  AUXILIARY LEMMAS

**Lemma 17** (Variant of Lemma 10 in Cheng & Bartlett (2018))**.** *Suppose* $-\log p_*$ *is* $m$-*strongly convex function, for any distribution with density function* $p$*, we have*

$$
\mathrm{KL}\left(p \big\| p_*\right) \le \frac{1}{2m} \int p(\boldsymbol{x}) \left\| \nabla \log \frac{p(\boldsymbol{x})}{p_*(\boldsymbol{x})} \right\|^2 \mathrm{d}\boldsymbol{x}.
$$

*By choosing* $p(\boldsymbol{x}) = g^2(\boldsymbol{x}) p_*(\boldsymbol{x}) / \mathbb{E}_{p_*}\left[g^2(\mathbf{x})\right]$ *for the test function* $g \colon \mathbb{R}^d \to \mathbb{R}$ *and* $\mathbb{E}_{p_*}\left[g^2(\mathbf{x})\right] < \infty$, *we have*

$$
\mathbb{E}_{p_*}\left[g^2 \log g^2\right] - \mathbb{E}_{p_*}\left[g^2\right] \log \mathbb{E}_{p_*}\left[g^2\right] \le \frac{2}{m} \mathbb{E}_{p_*}\left[\|\nabla g\|^2\right],
$$

*which implies* $p_*$ *satisfies* $m$-*log-Sobolev inequality.*

**Lemma 18** (Lemma 11 in Vempala & Wibisono (2019))**.** *Suppose a density function* $p \propto \exp(-f)$ *and* $\|\nabla^2 f\| \le L$*. Then, it has*

$$
\mathbb{E}_{\mathbf{x} \sim p}\left[\|\nabla f(\mathbf{x})\|^2\right] \le Ld
$$

*where* $d$ *is the dimension number of* $\boldsymbol{x}$*.*

**Lemma 19** (Lemma B.1 in Huang et al. (2024))**.** *Consider four random variables,* $\mathbf{x}, \mathbf{z}, \tilde{\mathbf{x}}, \tilde{\mathbf{z}}$*, whose underlying distributions are denoted as* $p_x, p_z, q_x, q_z$*. Suppose* $p_{x,z}$ *and* $q_{x,z}$ *denotes the densities of joint distributions of* $(\mathbf{x}, \mathbf{z})$ *and* $(\tilde{\mathbf{x}}, \tilde{\mathbf{z}})$*, which we write in terms of the conditionals and marginals as*

$$
p_{x,z}(\boldsymbol{x}, \boldsymbol{z}) = p_{x|z}(\boldsymbol{x}|\boldsymbol{z}) \cdot p_z(\boldsymbol{z}) = p_{z|x}(\boldsymbol{z}|\boldsymbol{x}) \cdot p_x(\boldsymbol{x})
$$
$$
q_{x,z}(\boldsymbol{x}, \boldsymbol{z}) = q_{x|z}(\boldsymbol{x}|\boldsymbol{z}) \cdot q_z(\boldsymbol{z}) = q_{z|x}(\boldsymbol{z}|\boldsymbol{x}) \cdot q_x(\boldsymbol{x}).
$$

*then we have*

$$
\mathrm{TV}\left(p_{x,z}, q_{x,z}\right) \le \min \Big\{ \mathrm{TV}\left(p_z, q_z\right) + \mathbb{E}_{\mathbf{z} \sim p_z}\left[\mathrm{TV}\left(p_{x|z}(\cdot|\mathbf{z}), q_{x|z}(\cdot|\mathbf{z})\right)\right],
$$
$$
\mathrm{TV}\left(p_x, q_x\right) + \mathbb{E}_{\mathbf{x} \sim p_x}\left[\mathrm{TV}\left(p_{z|x}(\cdot|\mathbf{x}), q_{z|x}(\cdot|\mathbf{x})\right)\right] \Big\}.
$$

*Besides, we have*

$$
\mathrm{TV}\left(p_x, q_x\right) \le \mathrm{TV}\left(p_{x,z}, q_{x,z}\right).
$$

**Lemma 20** (Lemma B.4 in Huang et al. (2024))**.** *Consider four random variables,* $\mathbf{x}, \mathbf{z}, \tilde{\mathbf{x}}, \tilde{\mathbf{z}}$*, whose underlying distributions are denoted as* $p_x, p_z, q_x, q_z$*. Suppose* $p_{x,z}$ *and* $q_{x,z}$ *denotes the densities of joint distributions of* $(\mathbf{x}, \mathbf{z})$ *and* $(\tilde{\mathbf{x}}, \tilde{\mathbf{z}})$*, which we write in terms of the conditionals and marginals as*

$$
p_{x,z}(\boldsymbol{x}, \boldsymbol{z}) = p_{x|z}(\boldsymbol{x}|\boldsymbol{z}) \cdot p_z(\boldsymbol{z}) = p_{z|x}(\boldsymbol{z}|\boldsymbol{x}) \cdot p_x(\boldsymbol{x})
$$
$$
q_{x,z}(\boldsymbol{x}, \boldsymbol{z}) = q_{x|z}(\boldsymbol{x}|\boldsymbol{z}) \cdot q_z(\boldsymbol{z}) = q_{z|x}(\boldsymbol{z}|\boldsymbol{x}) \cdot q_x(\boldsymbol{x}).
$$

*then we have*

$$
\mathrm{KL}\left(p_{x,z} \big\| q_{x,z}\right) = \mathrm{KL}\left(p_z \big\| q_z\right) + \mathbb{E}_{\mathbf{z} \sim p_z}\left[\mathrm{KL}\left(p_{x|z}(\cdot|\mathbf{z}) \big\| q_{x|z}(\cdot|\mathbf{z})\right)\right]
$$
$$
= \mathrm{KL}\left(p_x \big\| q_x\right) + \mathbb{E}_{\mathbf{x} \sim p_x}\left[\mathrm{KL}\left(p_{z|x}(\cdot|\mathbf{x}) \big\| q_{z|x}(\cdot|\mathbf{x})\right)\right]
$$

*where the latter equation implies*

$$
\mathrm{KL}\left(p_x \big\| q_x\right) \le \mathrm{KL}\left(p_{x,z} \big\| q_{x,z}\right).
$$

**Lemma 21** (Lemma C.1 in Chen et al. (2023a)). *Consider the following two Itô processes*

$$dX_t = F_1(X_t, t)\, dt + g(t)\, dW_t, \quad X_0 = a,$$
$$dY_t = F_2(Y_t, t)\, dt + g(t)\, dW_t, \quad Y_0 = a, \tag{59}$$

*where $F_1, F_2, g$ are continuous functions and may depend on $a$. We assume the uniqueness and regularity condition:*

- *The two SDEs have unique solutions.*

- *$X_t, Y_t$ admit densities $p_t, q_t \in C^2(\mathbb{R}^d)$ for $t > 0$.*

*Define the relative Fisher information between $p_t$ and $q_t$ by*

$$J(p_t \| q_t) = \int p_t(x) \left\| \nabla \log \frac{p_t(x)}{q_t(x)} \right\|^2 dx. \tag{60}$$

*Then for any $t > 0$, the evolution of $\mathrm{KL}\left(p_t \| q_t\right)$ is given by*

$$\frac{\partial}{\partial t} \mathrm{KL}\left(p_t \| q_t\right) = -\frac{g(t)^2}{2} J(p_t \| q_t) + \mathbb{E}\left[\left\langle F_1(X_t, t) - F_2(X_t, t), \nabla \log \frac{p_t(X_t)}{q_t(X_t)} \right\rangle\right]. \tag{61}$$

**Lemma 22** (Lemma C.2 in Chen et al. (2023a)). *For $0 \leq k \leq N - 1$, consider the reverse SDE starting from $\tilde{x}_{t'_k} = a$*

$$d\tilde{x}_t = \left[\frac{1}{2}\tilde{x}_t + \nabla \log \tilde{p}_t(\tilde{x}_t)\right] dt + dW_t, \qquad \tilde{x}_{t'_k} = a \tag{62}$$

*and its discrete approximation:*

$$d\hat{y}_t = \left[\frac{1}{2}\hat{y}_t + s(a, t - t'_k)\right] dt + dW_t, \qquad \hat{y}_{t'_k} = a \tag{63}$$

*for time $t \in (t'_k, t'_{k+1}]$. Let $\tilde{p}_{t|t'_k}$ be the density of $\tilde{x}_t$ given $\tilde{x}_{t'_k}$ and $\hat{q}_{t|t'_k}$ be density of $\hat{y}_t$ given $\hat{y}_{t'_k}$. Then we have*

1. *For any $a \in \mathbb{R}^d$, the two processes satisfy the uniqueness and regularity condition stated in Lemma 21, that is, 62 and 63 have unique solution and $\tilde{p}_{t|t'_k}(\cdot|a), \hat{q}_{t|t'_k}(\cdot|a) \in C^2(\mathbb{R}^d)$ for $t > t'_k$.*

2. *For a.e. $a \in \mathbb{R}^d$ (with respect to the Lebesgue measure), we have*

$$\lim_{t \to t'_k{}^+} \mathrm{KL}\left(\tilde{p}_{t|t'_k}(\cdot|a) \big\| \hat{q}_{t|t'_k}(\cdot|a)\right) = 0. \tag{64}$$

**Lemma 23** (Lemma C.9 in Chen et al. (2023a)). *Suppose that Assumption 3 holds. If $\sigma_t^2 \leq \frac{\alpha_t}{2L}$, we have $\nabla \log p_t$ is $2L\alpha_t^{-1}$-Lipschitz on $\mathbb{R}^d$.*

**Lemma 24** (Lemma C.6 in Chen et al. (2023a)). *For any $0 \leq t \leq s \leq T$, the forward process 3 satisfies,*

$$\mathbb{E}\left\|\nabla \log q_t(x_t) - \nabla \log q_s(x_s)\right\|^2 \leq 4\mathbb{E}\left\|\nabla \log q_t(x_t) - \nabla \log q_t(\alpha_{t,s}^{-1}x_s)\right\|^2 \tag{65}$$

$$+ 2\mathbb{E}\left\|\nabla \log q_t(x_t)\right\|^2 \left(1 - \alpha_{t,s}^{-1}\right)^2. \tag{66}$$

**Lemma 25** (Lemma 1 in Benton et al. (2024)). *For all $t > 0$, $\frac{\sigma_t^3}{2\dot{\sigma}_t} \frac{d}{dt}\mathbb{E}[\mathbf{\Sigma}_t] = \mathbb{E}[\mathbf{\Sigma}_t^2]$.*

## E   MORE DETAILS ON EXPERIMENTS

In this section, we add more detailed experimental settings omitted in Section 5.

## E.1 ADDITIONAL DETAILS OF CONDITIONAL KL COMPARISON EXPERIMENTS.

**Setup and evaluation.** For each task, we synthesize 3000 samples and use them to train both AR diffusion and DDPM. To ensure a fair comparison, both AR diffusion and DDPM are trained directly in pixel space without a VAE, and are configured identically with the same backbone (U-Net), total parameter count, and a training schedule of 500 epochs. To ensure a fair comparison, both AR diffusion and DDPM are trained directly in pixel space without a VAE. We adopt a unified learning rate of 3e-4, and both models have approximately 23M parameters. Training is conducted for 500 epochs on a single NVIDIA A800 GPU.

In the evaluation phase, we first pretrain a CNN on the MNIST dataset, achieving a test accuracy of 99.23%. For the generated data, we first divide the image into four sub-images along the vertical and horizontal midlines. Then, we use the pretrained CNN to predict the labels of the sub-images. We evaluate whether the sub-image labels satisfy the predefined dependencies, i.e., formning arithmetic sequences. If the predefined dependencies are satisfied, the sample is labeled as a correct sample.

## E.2 ADDITIONAL DETAILS OF CONDITIONAL DATA CAPTURING EXPERIMENTS.

**Setup and evaluation.** For each task, we synthesize 2000 samples and use them to train both AR Diffusion and DDPM. We follow the same experimental setup as Appendix E.1 such as both AR diffusion and DDPM are trained in pixel space. Training is carried out on a single NVIDIA A800 GPU for 200 epochs (Task 1) and 600 epochs (Task 2), respectively.

For the generated images, we can directly obtain image masks using predefined colors (e.g., red or blue for the square and rectangle in Task 1) during image generation. Using these masks, we can directly extract the variables of interest, such as the square's side length $l_1$ and the rectangle's side length $l_2$, and calculate their ratio $R = \frac{l_2}{l_1}$. For Task 1, the target ratio is $R = 1.5$, while for Task 2, the target ratio is $R = 5$.

# F THE USE OF LARGE LANGUAGE MODELS (LLMS)

In writing, we used LLMs for grammar checking and sentence polishing.

