# OpenReview forum: "Theory of Autoregressive Diffusion Model: Inference Complexity and Conditional Dependency Learning"
_ICLR.cc/2026/Conference — ICLR 2026 Conference Withdrawn Submission_

### Official Review · Reviewer_jHZv · 2025-10-16

**Soundness:** 3
**Presentation:** 2
**Contribution:** 1
**Rating:** 2
**Confidence:** 4

**Summary:**

This paper studies AR diffusion, which reverses a patch-wise diffusion model with conditioning from already sampled patches. The authors establish an error bound for AR diffusion models, and discusses the conditions on which AR diffusions outperform conventional diffusion models. They also empirically compare AR and conventional diffusion models through two toy examples, confirming that a certain conditional structure can be better learned by AR diffusions.

**Strengths:**

The paper is well written, and the mathematical derivations are easy to follow. Theorem 1 decomposes the error/complexity into $K$-related term (coming from $K$-stage AR formulation) and standard term (coming from usual diffusion error estimates), which helps the understanding of AR diffusion modeling. The toy examples also confirm the intuition that the AR formulation explicitly learning conditional distributions can be helpful when the patch decomposition aligns with the data-generation process.

**Weaknesses:**

A large portion of the paper (pages 1-6) is just rewriting the standard formulations of diffusion models (such as Chen et al. (2023b)) into AR-based formulations. The essential contributions of the paper seems to be the results in Section 4, but it has the following weaknesses:
- [W1] The complexity obtained in Theorem 1 (and its remark) seems to be just $K\times$ complexity of standard diffusion. Technically, it seems to be just a combination of stage-wise estimates of standard diffusion models. Furthermore, while the authors state "Moreover, we prove that, in certain regimes, AR diffusion can better capture some conditional dependencies among patches compared to standard diffusion" in Introduction, the "regime" is described only as $\epsilon(K)\ll K^{1/2}\epsilon(1)$, and it is far from characterizing the situation (in terms of data and/or network) where AR diffusion is more efficient.
- [W2] **The results in Section 4.2 is quite misleading.** First, for Lemma 4, the conditional KL divergence is bounded by using the conditional loss $L^{SM}(\theta|x_{[1:k]})$ and the authors state it can be arbitrarily small. However, in the loss functions such as Eq. 14, this conditional loss is taken expectation regarding the choice of $x_{[1:k]}$, and there is no reason that $L^{SM}(\theta|x_{[1:k]})$ is uniformly small for each $x_{[1:k]}$. Similarly, in Lemma 5, the right-hand side can be large when $\lVert x_{1:k}\rVert$ is moderately large, but in the example constructed by authors (in Appendix C), $\lVert x_{1:k}\rVert$ **tends to be very small**, since it follows a centered Gaussian with variance of order $O(\epsilon^2/M^2)$. Thus, "unbounded KL divergence" stated by the authors in Introduction is an inappropriate exaggeration of the result. Indeed, from the chain rule of KL divergence, the conditional KL divergence is small in expectation, if the joint KL divergence is small.

Overall, I believe that the paper's theoretical contribution is only in Section 4.1, which gives a bound for AR diffusion models. While the result follows the intuition, it does not explain the competence of AR diffusions in a certain setting. I do not think the additional empirical findings in Section 5 are strong enough to get the paper accepted.

Typos etc:
- L56: learns -> learn
- L87: "regimes ," -> "regimes,"
- L129: a coefficient (i.e., 2) of $s_\theta$ is missing
- L138: What is $\theta_{df+1, k}$? Is it $\theta_{dm, k+1}$?
- L158: often often -> often
- L328: "$\delta\leq$" -> "$\delta\in$"
- L355: $K^{1/2}$ -> $K^{-1/2}$?
- L382: "$k+1|[1:k]$" is missing from $\hat{p}_*$

**Questions:**

- In which situations do we have $\epsilon(K)\ll K^{1/2}\epsilon(1)$? (please see [W1] as well)

---

### Official Review · Reviewer_S3Fh · 2025-10-29

**Soundness:** 1
**Presentation:** 2
**Contribution:** 2
**Rating:** 2
**Confidence:** 3

**Summary:**

This paper provides a formal theoretical analysis of the inference complexity for auto-regressive (AR) diffusion models. The authors begin by formalizing the AR inference process as a series of stage-wise conditional distribution samplings.

The central contribution is a derivation of the sampling complexity required to achieve a target KL divergence between the true data distribution and the one generated by the AR diffusion. The authors show this complexity to be $O(Kd\epsilon^{-2})$, where $K$ is the number of stages, $d$ is the dimension, and $\epsilon$ is the average training error across all stages.

Furthermore, the authors prove a key contrasting point: while standard (vanilla) diffusion models can effectively learn accurate joint distributions, their estimates of conditional distributions can, in fact, be arbitrarily biased.

**Strengths:**

* **Clarity of Exposition:** The paper is well-written and logically structured. The authors begin with a detailed introduction and a **rigorous formulation** of auto-regressive diffusion models, which effectively **lays the groundwork** for the subsequent theoretical analysis.


* **Grounded Theoretical Analysis:** The analysis is **grounded in established assumptions** from prior work (e.g., minimal smoothness requirements), although it does introduce an **additional assumption** of an upper bound on the gradient norm.

**Weaknesses:**

* **Lack of Analysis on Error Accumulation:** A primary weakness is the paper's failure to address the well-known problem of **error accumulation** in stage-wise generative processes. It is established that in AR models, sampling errors from one stage can propagate and compound, significantly degrading the quality of subsequent generations. This is especially critical for long-horizon tasks like video generation. The paper lacks a formal analysis of this effect, which **significantly undermines the soundness** of its complexity claims, as the derived bounds may not hold in a practical, recursive sampling setting.

* **Apparent Flaw in the KL Decomposition (Line 916):** The aforementioned lack of error analysis may stem from a specific flaw in the KL decomposition (Line 916). The derivation appears to **incorrectly condition on ground-truth samples** from the previous stage ($x_{k-1}$) instead of the **model's own generated samples** ($\tilde{x}_{k-1}$). This is a critical distinction: the analysis seems to be based on a "teacher-forced" setting, not the true, auto-regressive inference process where errors would naturally accumulate.

* **Questionable Smoothness Assumptions:** The paper's theoretical results hinge on smoothness assumptions that are likely **violated in many practical AR settings**. For example, in video generation, the objects' motion follows physical laws, making the conditional distribution of the next frame given the previous one **nearly deterministic (i.e., approximating a delta distribution)**. For such sharp distributions, the smoothness assumptions no longer hold. Could the authors provide some analysis or discussion on these non-smooth cases?

**Questions:**

Please see weakness

---

### Official Review · Reviewer_6aas · 2025-10-30

**Soundness:** 3
**Presentation:** 3
**Contribution:** 2
**Rating:** 4
**Confidence:** 4

**Summary:**

This paper provides the first theoretical analysis of the sampling error for autoregressive (AR) diffusion models. The authors formalize the AR inference process as a series of stage-wise conditional distribution samplings and derive corresponding inference complexity bounds.

**Strengths:**

The paper addresses an important and previously unexplored theoretical question: the inference complexity of autoregressive diffusion models. The problem setting is clearly articulated, and the paper is generally well-written. To the best of my knowledge, this is the first work to provide a formal analysis of sampling error in this specific AR context.

**Weaknesses:**

Despite its clarity, the paper's contribution appears to be limited due to its incremental nature, reliance on strong assumptions, and a failure to engage with the current state-of-the-art in diffusion model theory.
1. **Incremental Contribution**: The core technical analysis seems to be a straightforward extension of prior work on standard (non-AR) diffusion models, specifically [Chen et al., 2022](https://arxiv.org/abs/2209.11215). As noted in the draft, when the analysis is specialized to the non-AR case ($K=1$, $T \approx 1$), the resulting sampling error $\tilde{O}(L^2d \eta)$ appears to be the same as the bound in [Chen et al., 2022](https://arxiv.org/abs/2209.11215). This seems to suggest that the AR framework does not introduce fundamental technical challenges that necessitate a new analysis framework.
2. **Out-of-Date Comparisons**: As mentioned in 1, the paper's theoretical bounds are benchmarked against the [Chen et al., 2022] result. This comparison is severely out-of-date, as the field has advanced significantly. Recent works have established much tighter convergence guarantees, often under weaker assumptions. For example, [Li et al., 2024](https://arxiv.org/pdf/2409.18959), establishes an $O(d/T)$ convergence rate in total variation distance under minimal assumptions. The submission completely lacks a discussion of these state-of-the-art results, making it impossible to assess the significance of its own bounds.
3. **Strong and Unusual Assumptions**: The analysis requires an upper bound on the Hessian norm (in Assumption A2). This is a strong condition that is not standard in related literature. For instance, [Chen et al., 2022](https://arxiv.org/abs/2209.11215) only requires an L-Lipschitz score function and finite second moments. The paper fails to provide a clear justification for why this much stronger assumption is necessary for the AR setting.

**Questions:**

1. Could the authors please clarify the necessity of the Hessian norm upper bound in Assumption A2? Why do standard assumptions (e.g., L-Lipschitz score) suffice for non-AR models, but this stronger condition is required for the AR analysis?
2. What is the primary technical difficulty or novelty introduced by the autoregressive setting? The proof techniques seem to closely mirror those in [Chen et al., 2022](https://arxiv.org/abs/2209.11215), and the resulting bound for $K=1$ is identical. A clearer articulation of the novel technical contribution is needed.
3. The paper's results are benchmarked against [Chen et al., 2022](https://arxiv.org/abs/2209.11215). Could the authors provide a comprehensive discussion and comparison of their bounds with more recent, state-of-the-art results, such as those in [Chen et al., 2023](https://proceedings.mlr.press/v202/chen23q/chen23q.pdf) or the $O(d/T)$ bounds in [Li & Yan, 2024](https://arxiv.org/pdf/2409.18959)? Contextualizing the work within these recent advancements is crucial.
4. The paper's analysis is based on a "sequential patch-by-patch" factorization. This setup seems to differ from successful practical models (e.g., [VAR](https://arxiv.org/pdf/2404.02905)) that rely on a "coarse-to-fine 'next-scale prediction'" or upscaling structure. Could the authors elaborate on the connection between their theoretical framework and these hierarchical, multi-scale architectures? Can we have more fine-grained analysis hold for models where the "stages" are closely coupled as different resolutions?

---

### Official Review · Reviewer_Ved4 · 2025-11-01

**Soundness:** 3
**Presentation:** 3
**Contribution:** 3
**Rating:** 6
**Confidence:** 1

**Summary:**

This paper provides a theoretical investigation of autoregressive (AR) diffusion models, a class of generative models that perform patch-by-patch conditional sampling. The authors derive complexity bounds for AR diffusion inference under mild smoothness assumptions, showing that its sequential structure introduces an additional factor related to the number of stages. They further demonstrate that this stage-wise design can better capture conditional dependencies often missed by standard diffusion models, with supporting validation on synthetic experiments.

**Strengths:**

* This work is among the first to provide theoretical analysis of autoregressive (AR) diffusion models, clarifying their inference complexity and conditional dependency learning, an area previously dominated by empirical progress without solid theoretical grounding.

* The paper derives complexity bounds for AR diffusion inference under mild smoothness and boundedness assumptions, offering formal convergence guarantees and extending the theoretical framework used in standard diffusion models.

* Beyond global distribution matching, the analysis highlights how AR diffusion’s stage-wise conditional formulation enables it to better capture local or structured dependencies (e.g., between image patches), a property supported by accompanying synthetic experiments.

**Weaknesses:**

* While the theoretical results seem to be solid, I do have concerns regarding the empirical results of this paper. Most experiment results are limited to a 4$\times$4 image grid with MINIST data, which makes the evaluation fall into a toy example. There is no experiments on relatively large-scale datasets or standard benchmark. Additionally, video generation might be the one that suits the AR diffusion best. This raises serious concerns about scalability and practical relevance, can theoretical insights remain untested in settings where diffusion models are actually deployed.

**Questions:**

* While autoregressive diffusion models show promising theoretical properties, their adoption in large-scale image and video generation remains limited. Could the authors discuss the current practical challenges or limitations that hinder the broader application of AR diffusion in these domains?

* I am not an expert in this field, so I was struggling to follow the math in this paper at most time. But I will try to check other reviewers' comment and engage in the discussion.

---

### Note · Authors · 2025-11-13

I have read and agree with the venue's withdrawal policy on behalf of myself and my co-authors.